# Mechanistic and Compositional Aspects of Industrial Catalysts for Selective CO$_2$ Hydrogenation Processes

**Guido Busca** [1,2,*] , **Elena Spennati** [1,2] , **Paola Riani** [2,3] and **Gabriella Garbarino** [1,2]

1    Department of Civil, Chemical and Environmental Engineering, University of Genoa, Via Opera Pia 15,
     16145 Genova, Italy; Elena.Spennati@edu.unige.it (E.S.); gabriella.garbarino@unige.it (G.G.)
2    Research Unit of Genoa, National Inter-University Consortium of Science and Technology of Materials (INSTM),
     Via Dodecaneso 33, 16146 Genova, Italy; paola.riani@unige.it
3    Department of Chemistry and Industrial Chemistry, University of Genoa, Via Dodecaneso 33,
     16146 Genova, Italy
*    Correspondence: guido.busca@unige.it; Tel.: +39-010-335-6024

**Abstract:** The characteristics of industrial catalysts for conventional water-gas shifts, methanol syntheses, methanation, and Fischer-Tropsch syntheses starting from syngases are reviewed and discussed. The information about catalysts under industrial development for the hydrogenation of captured CO$_2$ is also reported and considered. In particular, the development of catalysts for reverse water-gas shifts, CO$_2$ to methanol, CO$_2$-methanation, and CO$_2$-Fischer-Tropsch is analyzed. The difference between conventional catalysts and those needed for pure CO$_2$ conversion is discussed. The surface chemistry of metals, oxides, and carbides involved in this field, in relation to the adsorption of hydrogen, CO, and CO$_2$, is also briefly reviewed and critically discussed. The mechanistic aspects of the involved reactions and details on catalysts' composition and structure are critically considered and analyzed.

**Keywords:** carbon dioxide; hydrogenation; methanol synthesis; methanation; water-gas shift; Fischer-Tropsch synthesis; copper; nickel; iron; cobalt

## 1. Introduction

E-fuels are, by definition, burnable compounds that are (or, better, will be) produced from captured CO$_2$ and green electrolytic hydrogen [1]. These fuels can be used in conventional internal combustion engines, thus reducing emissions of greenhouse gases of fossil origins, which are the likely causes of global warming. CO$_2$ hydrogenation [2] can also produce, in a renewable way, compounds, e.g., olefins, for application as chemical intermediates in the field of a future green chemical industry.

Most of the compounds that could be produced in the future via CO$_2$ hydrogenation are manufactured today, starting from fossil-derived syngases, i.e., CO-rich CO$_x$ + H$_2$ mixtures produced mostly either from steam reforming of natural gas [3,4], or from coal gasification [5]. Relevant data on these syngas-based processes and the respective industrial catalysts are reported in Table 1. Such compounds could also be produced, in principle, starting from biomass-derived syngases [6] as biofuels or renewable chemicals. To convert these processes from CO-rich feeds to pure CO$_2$ feedstocks, both catalysts and process modification may be needed.

**Table 1.** Catalysts and conditions of current commercial processes for the conversion of syngases (for references, see the text).

| Process | | Main Component | | Other Components | | | P Range | T Range | Feed Composition | | | | |
|---|---|---|---|---|---|---|---|---|---|---|---|---|---|
| | | Phase | wt% | Comp. | Role | wt% | Bar | K | CO mol% | $H_2$ mol% | $CO_2$ mol% | $H_2O$ mol% | Other mol% |
| WGS | HT, high SR [a] | $Fe_3O_4$ | 70–90 | $Cr_2O_3$, | stabilizer | <10 | 30–50 | 650–720 | 5–10 | 30–40 | 3–8 | 30–40 | $N_2$ < 15 |
| | | | | Cu | promoter | <3 | | | | | | | $CH_4$ < 0.5 |
| | | | | MgO | promoter | <1 | | | | | | | Ar < 0.5 |
| | HT, low SR [a] | ZnO-$ZnAl_2O_4$ | ~100 | | | | | | 6–12 | 45–50 | 4–10 | 15–20 | $N_2$ < 18 |
| | | | | | | | | | | | | | $CH_4$ < 0.6 |
| | | | | | | | | | | | | | Ar < 0.6 |
| | LT | Cu | >40 | ZnO | Activity promoter, stabilizer | 50–30 | 30–50 | 450–573 | 2–2.5 | 35–50 | 9–15 | 20–40 | $N_2$ 12–16 |
| | | | | $Al_2O_3$ (in ZnO) | stabilizer | <20 | | | | | | | $CH_4$ < 0.5 |
| | | | | $Cs_2O$, $Na_2O$ | Selectivity promoter | <1 | | | | | | | Ar < 0.5 |
| Methanol synthesis | | Cu | >50 | ZnO | Activity promoter, stabilizer | ~40 | 50–150 | 473–523 | 10–35 | 40–75 | 1–13 | <2 | $CH_4$ < 15 |
| | | | | $Al_2O_3$ (in ZnO) | stabilizer | <10 | | | | | | | |
| | | | | MgO | promoter | ~2 | | | | | | | |
| | | | | $SiO_2$ | stabilizer | | | | | | | | |
| Methanation | LT [a] | Ru | 0.3 | $\gamma$-$Al_2O_3$ | support | 99 | 30–50 | 440–550 | 0.5 | 75 | 0.2 | --- | $N_2$ 24 |
| | | Ni | 20–50 | $\gamma$-$Al_2O_3$ | support | <80 | 30–50 | 470–620 | | | | | $CH_4$ 0.2 |
| | | | | MgO, CaO, $La_2O_3$ | stabilizer | <20 | | | | | | | Ar 0.3 |
| | HT | Ni | >20 | $MgAl_2O_4$ $La_2O_3$-$Al_2O_3$ CaO-$Al_2O_3$ | support, stabilizer | <80 | >30 | 500–970 | 25–35 | 35–75 | 1–30 | --- | $CH_4$ < 10 |
| FTS | LT | Co | 15–30 | $\gamma$-$Al_2O_3$ | support | <80 | 20–30 | 473–523 | 25–45 | 45–70 | 0–5 | <1 | $N_2$ < 10 $CH_4$ < 10 |
| | | | | Ru, Rh, Pt or Pd | activator | <0.1 | | | | | | | |
| | | | | $ZrO_2$, $CeO_2$, $La_2O_3$ | promoter, stabilizer | <10 | | | | | | | |
| | | | | $CoC_2$ | Inert [b] | <6 | | | | | | | |
| | HT | Fe | 90 | $SiO_2$ | promoter | <5 | 20–40 | 590–630 | 35–60 | 35–60 | 0–5 | <1 | $N_2$ < 10 $CH_4$ < 10 |
| | | | | Cu | promoter | <5 | | | | | | | |
| | | | | $K_2O$ | promoter | <5 | | | | | | | |
| | | | | Fe carbides | active phases [b] | | | | | | | | |
| | | | | $Fe_3O_4$ | | | | | | | | | |

[a] Data for the processes realized for the preparation of ammonia synthesis gas. [b] Phases formed under reaction.

As it is well known, transition metals are the most catalytically active materials for hydrogenation reactions, at least in sulfur-free environments [7]. In fact, the large majority of catalysts used industrially for hydrogenations are metal-based. However, most of them are supported on metal oxides or, at least, present in composites with metal oxides. This is also true for catalysts for the hydrogenations of both CO and $CO_2$, where, however, metal carbides can be present and act in catalysis. Thus, from the mechanistic point of view, several questions may arise concerning the origins of catalyst activity and selectivity towards one or another product and the role of supports and different catalyst components in a catalyst's behavior. These points may be relevant also in developing new catalysts for $CO_2$ hydrogenation technologies.

As it will become clear from our review, and is a common fact in industrial catalysis, although an enormous number of different catalyst compositions are objects of investigation in academic research, as well as in applied industrial research, industries converge usually over very few (frequently only one) main catalyst compositions that are found to be most performant and stable in industrial practice. In fact, commercial catalysts offered by different producers for the same reaction frequently differ only for small amounts of promoters or slight morphological differences, being based on the same chemical system. For example, as reported by Gao et al. [8] in 2015 and by Rönsch et al. [9] in 2016, detailed academic studies on $CO_x$ methanation catalysts concerned at least eleven different metals as active phases (Ru, Ir, Rh, Ni, Co, Os, Pt, Fe, Mo, Pd, Ag), unsupported or supported on at least eight different carrier families ($Al_2O_3$, $SiO_2$, $TiO_2$, $ZrO_2$, $CeO_2$, SiC, hexa-aluminates, perovskites), as well as a number of promoters, leading to hundreds of combinations. In contrast, probably as a result of these research efforts, only two systems were applied commercially for this reaction, i.e., $Ru/Al_2O_3$- and $Ni/Al_2O_3$-based catalysts (see below).

On the other hand, it is also clear that the development of the optimal catalyst is the key to the design of the reactor(s) and product purification sections, i.e., for the development of the entire new process. Indeed, catalyst manufacturing is a key activity in the overall chemical industry: the global catalyst market size was valued at USD 29.7 billion in 2022 and is anticipated to grow at a compound annual growth rate (CAGR) of 4.6% from 2023 to 2030 [10].

In the present paper, an analysis of the state of the art of this technological transition, from syngas conversion to $CO_2$ hydrogenation, is attempted, and related surface chemical and mechanistic aspects are discussed.

## 2. The Water-Gas Shift and Reverse Water-Gas Shift Equilibrium and the Catalysts

In Figure 1, the standard free energy of relevant reactions ($\Delta G°$) as functions of temperature are reported for both CO (left) and $CO_2$ (right) conversion. The water-gas shift (WGS) reaction and its reverse (rWGS), as well as methanation and methanol synthesis reactions, are considered, with the addition of dimethylether synthesis. The last reaction, which will not be further discussed in the present review, produces an important intermediate in the frame of methanol to chemical processes (methanol to olefins, MTO, as well as methanol to gasoline, MTG), of which its role may grow in the frame of the expected energy transition. In all cases, CO-based reactions are more favored than $CO_2$-based ones. Additionally, $CO_x$ hydrogenation to methane is always the most favored below ~900 K, while above this temperature, WGS and rWGS, in CO and $CO_2$ cases, respectively, become the least favored in the considered situation.

The water-gas shift (WGS) equilibrium [11] is constituted of two opposite reversible reactions that allow for the modification of the $CO_x/H_2$ ratio in syngas and to convert CO into $CO_2$ or *vice versa*.

$$CO + H_2O \rightleftharpoons CO_2 + H_2 \qquad \Delta H°_{298} = -41.2 \ kJ/mol \qquad \Delta G°_{298} = -28.7 \ kJ/mol \qquad (1)$$

The reaction is weakly sensitive to pressure. According to the exothermicity of the direct WGS reaction, it can be realized at low temperatures. The $\Delta G°$ value increases with temperature and becomes positive at temperatures around 1098 K, above which

the reverse reaction rWGS becomes more favored [12]. In the most common case today, the WGS process is applied to syngas produced from the steam reforming of natural gas to convert CO into $CO_2$ (less toxic, more useful, more easily separated from $H_2$, and more easily managed) and to further produce $H_2$ in processes aimed at the production of hydrogen or ammonia synthesis gases. The WGS process is frequently realized in two steps, high-temperature WGS and low-temperature WGS, as usual for ammonia synthesis gas production [3,13], or in a single step at a high [14] or medium temperature [15] if a pressure swing adsorption (PSA) step follows to deeply purify the produced hydrogen.

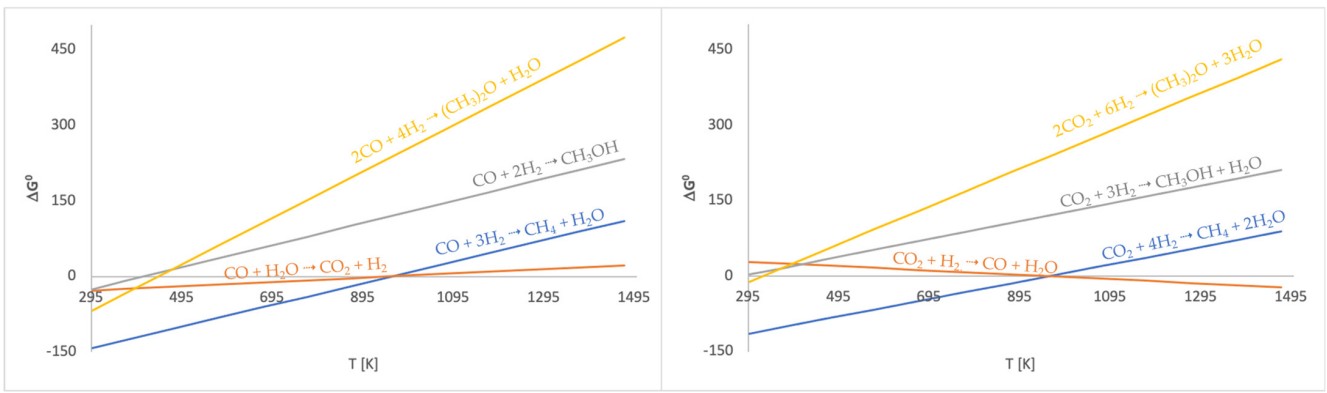

**Figure 1.** Standard Gibbs free energy vs. T for selected CO (**left**) and $CO_2$ (**right**) reactions.

## 2.1. High-Temperature Water-Gas Shift (HTWGS) Catalysts

High-temperature water-gas shift is usually carried out at 650–720 K, decreasing the CO concentration in syngas down to 1–3% mol on a dry basis [3,16]. Due to the limited exothermicity of the reaction, a single fixed-bed adiabatic reactor [13,14], or, sometimes, more adiabatic reactors with intermediate cooling, are used. Cheap and robust $Fe_2O_3$-based catalysts, frequently stabilized with $Cr_2O_3$ and also containing MgO, with a typical composition of 74.2% $Fe_2O_3$, 10.0% $Cr_2O_3$, and 0.2% MgO, and the remaining volatiles [11], are used. In the most recent formulation, 1–3% copper is added to the catalyst, improving catalytic activity [16]. The Clariant ShiftMax 120 catalyst is reported to have $Fe_2O_3 > 80\%$, $Cr_2O_3$ 8.5%, and Cu 2% [17], while the Johnson Matthey KATALCO 71-6 is based on $Fe_2O_3$ with ~2.5 wt% CuO and 9 wt% $Cr_2O_3$ [18]. Under reaction conditions, $Fe_2O_3$ (hematite/maghemite) is reduced to $Fe_3O_4$ (magnetite), which is stabilized morphologically and structurally by chromium, producing a spinel structure with a composition of $Fe[Fe_{2-x}Cr_x]O_4$, with a medium–low specific surface area (10–50 $m^2/g$). It has been concluded that, in this case, copper metal is mainly acting as an activator for iron oxide.

A high steam-to-dry gas ratio is needed in the feed to prevent iron carbide formation on these catalysts, resulting in unwanted by-product formation and reduced catalyst strength. Another limit of this catalytic system is associated with the pyrophoricity of the reduced working catalyst, thus needing a complex procedure for process shutdown. A further drawback is associated with the presence of chromium, which, upon the preparation of the catalyst, may result in the hexavalent state, characterized by high toxicity. For these reasons, Cr-free catalysts are under development [19,20]. The substitution of chromium with Al gives rise to lower activity catalysts. The addition of other elements, such as Zn, Ce, and Mn, is under study. In the case of the two-WGS-step process, the first HTWGS bed also works as a guard bed to protect the more sensitive and expensive catalyst used in the second LTWGS step.

To enlarge the operating steam-to-carbon ratio window (limited to avoid the formation of iron carbides in the case of iron oxide-based catalysts), Topsøe recently developed a new Cr- and Fe-free catalyst based on zinc oxide and zinc aluminum spinel, SK-501 Flex™, which is unique because of its stability under dry conditions [21].



### 2.2. Low-Temperature Water-Gas Shift (LTWGS) Catalysts

A low-temperature water-gas shift is carried out at 450–573 K using more active, but also more expensive and delicate, copper-based catalysts [16,22]. These catalysts need an absolute absence (<0.1 ppm) of sulfur and chlorine. Due to the small exothermicity of the reaction, a single adiabatic reactor is commonly used [23], although (pseudo)isothermal reactors have also been developed [15]. After the LTWGS reactor, CO concentration in syngas may be as low as 0.1% on a dry basis. The most relevant feature of these catalysts is high selectivity for WGS, reducing the coproduction of methanol, which will be recovered in the process condensate. The most recent LTWGS catalyst from Clariant ShiftMax$^{®®}$ 217 is based on 45% CuO, 44% ZnO, $Al_2O_3$ [17], and the addition of a special promoter [24] to limit methanol coproduction. It is used together with the guard catalyst SHIFTGUARD 200, based on 45% CuO, $Al_2O_3$, and a proprietary metal oxide, which effectively adsorbs and strongly retains chlorides so that the downstream LTWGS catalyst is protected [24]. Interestingly, the same company (Süd Chemie at that time) previously produced the same application a catalyst which was even richer in copper (ShiftMax$^{®®}$ 240, 57% CuO, 31% ZnO, 11% $Al_2O_3$, 1% promoter [25]).

The Topsøe LK-823 catalyst, as well as its latest version LK-853 FENCE™, are also based on Cu-ZnO-$Al_2O_3$ and are promoted by cesium to limit methanol formation [26,27]. Topsøe et al. also reported data on a guard catalyst constituted by Cu-ZnO-$Cr_2O_3$ (14% Cu) to protect copper/zinc/aluminum against chlorine poisoning [28]. Cu-ZnO-$Al_2O_3$ and alkali-promoted Cu-ZnO-$Al_2O_3$ are also sold by Johnson Matthey. According to these producers [29], alkali doping suppresses methanol formation and also boosts poison pick-up.

The HiFUEL$^{®®}$W220 LTWGS catalyst (sold by Alfa Aesar, Haverhill, MA, USA) contains CuO 52.5 wt%, ZnO 30.2 wt%, $Al_2O_3$ 17.0 wt%, and others at 0.3 wt% [30]. A commercial low-temperature shift catalyst, BASF K3-110, was reported to contain nominally 40% CuO, 40% ZnO, 20% $Al_2O_3$, BET area 102 $m^2$/g, pore volume 0.35 mL/g, copper area 9.83 $m^2$/g, dispersion 4.8%, and copper crystallite size 219 Å [31]. BASF later patented a new preparation method to produce water-gas shift catalysts that, in the unreduced form, have 50.5–52.3 wt% CuO, 27.1–29.6 wt% ZnO, 18.8–19.7 wt% $Al_2O_3$, and 0.08–0.1 wt% Na [32]. These catalysts too have pyrophoric behavior and must be carefully discharged to prevent risks.

It can be noted that LTWGS catalysts are quite similar to methanol synthesis catalysts, which are also based on Cu-ZnO-$Al_2O_3$ with quite similar component ratios (see below) [33,34]. Indeed, while reaction temperatures and $CO_x/H_2$ ratios can be also comparable in the two reactions, LTWGS is realized at a lower pressure and with much more water vapor in the reaction conditions than methanol synthesis. Thus, the real state of the catalyst surface in the two cases could differ significantly. In fact, more pronounced sintering of copper particles is reported to occur upon LTWGS than upon methanol synthesis [35]. In any case, metallic copper particles are the most abundant phase in these catalysts and cannot be considered as "supported" on the ZnO-$ZnAl_2O_4$ oxide minor component. Several Transmission electron microscopy studies [36–39] show that large copper particles are separated from each other by large ZnO-$ZnAl_2O_4$ particles, as schematized in Figure 2. However, data show that a strong interaction between substantially unsupported metallic copper and zinc species certainly occurs, leading to the formation of the most active species and modifying the surface of copper particles. This interaction can also involve metallic Zn species or Cu-Zn alloy [40], copper-supported ZnO layers [41], and/or partially oxidized Cu$^+$ sites [42]. The real nature of the most active species in these catalysts is still far from being clearly identified but certainly involves a surface modification of metallic copper by zinc species. Most of the Zn-Al oxide phase, constituted by ZnO where Al ions are present as spinel domains [43], act as structure stabilizer phases against sintering.

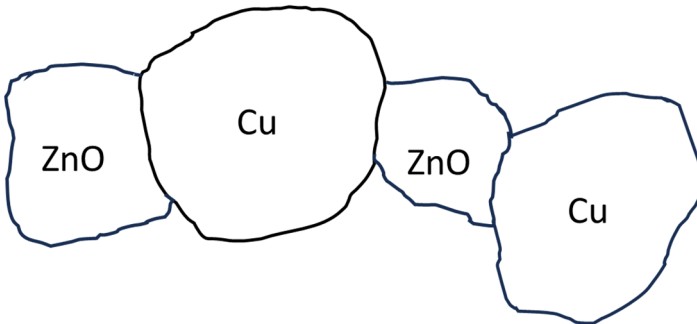

**Figure 2.** Schematics of TEM images of Cu/ZnO(Al$_2$O$_3$) catalysts. Al ions are in the ZnO phase, while the surface of copper particles is modified by additional Zn species.

### 2.3. Medium-Temperature Water-Gas Shift (MTWGS) Catalysts

When hydrogen purification is performed later by PSA, medium-temperature WGS (MTWGS) can be realized at about 490 to 550 K down to approx. 0.5% CO on a dry basis at the reactor outlet, realized either in adiabatic reactors or in isothermal water-cooled reactors [15]. Catalysts for MT-WGS are designed to be more thermally stable and manage exothermicity under low steam-to-carbon conditions [44]. Although nickel-based catalysts have also been proposed for MTWGS [45], it seems that industrial catalysts are also based on Cu/ZnO-Al$_2$O$_3$ [46]. In particular, Clariant's ShiftMax 300 catalyst (Clariant Produkte, Munich, Germany) is a stabilized copper–zinc catalyst reported to be ideally suited for the MTWGS [47]. In order to achieve optimum MTS performance, a composite loading, consisting of different types of Cu/ZnO-Al$_2$O$_3$-based MTS catalysts at the top and the bottom of the reactor can be used, as proposed by Topsøe, with the more active LK-813 catalyst on the top and the less active and more thermostable LK-819 catalyst at the bottom [48].

### 2.4. Other Industrial Water-Gas Shift Catalysts

Different versions of the WGS process can be applied and need different catalysts. Noble metal-based catalysts, such as Pt/CeO$_2$ [49], have even higher activity and can be preferentially used in miniaturized applications, e.g., for on-board hydrogen production, where more active catalysts and smaller reactors are needed.

The sour shift process includes WGS applied to S-containing gases coming from coal gasification. In this case, cobalt promoted MoS$_2$/$\gamma$-Al$_2$O$_3$-based catalysts, such as the SSK-10 catalyst from Topsøe, Lyngby, Denmark [50], and SHIFTMAX 820 and 821 from Clariant, Munich, Germany [17].

### 2.5. The Reverse Water-Gas Shift Process and Catalysts

The reverse water-gas shift reaction (rWGS) could be used to reduce the amount of hydrogen and CO$_2$ and increase that of CO in syngases with low H$_2$/CO$_x$ ratios (as those formed by coal gasification or by methane autothermal reforming). However, this was never realized, apparently, up to now: in fact, external CO$_2$ additions or recycling to syngas was a more convenient procedure to obtain the same result.

In contrast, the use of pure CO$_2$ feeds, such as captured CO$_2$, makes rWGS an interesting approach to produce CO or CO-containing syngases as the first step for further conversion [51], e.g., to synthetic fuels from the Fischer-Tropsch process, acetic acid, oxo-alcohols, or methanol, as discussed below. The reverse water-gas shift process is also a candidate technology for water and oxygen production on Mars under the in situ propellant production project [52] due to high ($\sim$95%) atmospheric CO$_2$ concentrations on Mars and the availability of H$_2$ as a byproduct of oxygen generation needed in these conditions.

Evidently, rWGS is an endothermic reaction which is shifted to products only above around 1100 K (Figure 1) when its $\Delta G°$ starts to become negative. As expected for the micro-reversibility principle, the same materials catalyzing WGS (such as Cu/ZnO-based LTWGS

catalysts) also catalyze rWGS at low temperatures [53]. Noble metal-based catalysts [54] and $Cu/CeO_2$ catalysts [55] are proposed to realize rWGS at medium temperatures (673–873 K). However, to push conversion towards CO and to obtain relatively low $H_2$ to CO ratios (e.g., for the Fischer-Tropsch reaction), rWGS must be realized at definitely higher temperatures (>1200 K), i.e., at temperatures where copper catalysts are unstable due to copper being near its melting point of 1257 K. In these conditions, several metal catalysts show activity for rWGS but with incomplete selectivity due to the coproduction of hydrocarbons and methanol [56], which can occur in small amounts, particularly at high pressures, together with the production of coke.

Interest in rWGS is now focusing on Ni-based catalysts, which are stable and commonly used at high temperatures. In particular, several projects were undertaken for the production of sustainable aviation fuel starting from $CO_2$ and electrolytic hydrogen, where rWGS is mostly realized using electrically heated reactors [57], such as the eREACT™ technology [58] used to realize the electrified reverse water-gas shift (eRWGS™), developed by Topsøe [59], in order to avoid $CO_2$ emissions. Using a feed of $H_2/CO_2$ in a ratio of 2.25 at 10 barg and high-temperature operation at 1320 K over a $Ni/ZrO_2$ washcoat catalyst, the production of synthesis gas with a $H_2/CO$ ratio of 2.0 and no detectable methane, ideal for Fischer-Tropsch synthesis, was obtained. The authors suggest that a sequential $CO_2$ methanation + methane steam reforming reaction scheme may occur. Also, Clariant developed a promoted nickel catalyst, ShiftMax®® 100 RE, for the rWGS shift as the first step in the production of e-fuels in collaboration with Ineratec, Karlsruhe, Germany [60]. According to Clariant, this catalyst shows high resistance against coking, low-methane byproduct formation, long lifetime, and high strength.

Johnson Matthey recently revealed the launch of HyCOgenTM [61], a reverse water-gas shift technology designed to help enable the conversion of captured $CO_2$ and green hydrogen into sustainable aviation fuel (SAF) through the conventional Fischer-Tropsch process (FT CANS technology [62]). Axens, Paul Wurth, and IFP Energies Nouvelles (IFPEN) have signed a co-development agreement for the optimization of the rWGS technology and its integration into e-fuel projects [63]. Indeed, the idea to develop rWGS as a first step to produce methanol from $CO_2 + H_2$ feeds was reported already several years ago [64].

## 3. The Methanol Synthesis and the Catalysts

The relevant reactions in methanol synthesis [65,66] are the following:

$$CO + 2\,H_2 \rightleftharpoons CH_3OH \quad \Delta H^\circ{}_{298} = -90.7 \text{ kJ/mol} \quad \Delta G^\circ{}_{298} = -25.3 \text{ kJ/mol} \tag{2}$$

$$CO_2 + 3\,H_2 \rightleftharpoons CH_3OH + H_2O \quad \Delta H^\circ{}_{298} = -49.5 \text{ kJ/mol} \quad \Delta G^\circ{}_{298} = +3.3 \text{ kJ/mol} \tag{3}$$

Also, the water gas shift equilibrium (Reaction (1)) may be established upon the reaction.

The methanol synthesis from CO (Reaction (1)) is favored ($\Delta G^\circ < 0$) at temperatures below 408 K. Taking into account that the best industrial catalysts have an activity threshold at about 450 K, the reaction is carried out in conditions where $\Delta G^\circ$ is positive. To shift to methanol equilibrium, high pressure is needed.

The methanol synthesis from $CO_2$ (Reaction (2)) is favored ($\Delta G^\circ < 0$) only at even lower temperatures than the synthesis from CO, i.e., below 280 K. Thus, thermodynamically, methanol synthesis from $CO_2$ hydrogenation is even less favored (lower C conversion in the same temperature and pressure conditions) than its synthesis from CO (Figure 1). Equilibrium becomes less and less favorable by increasing the $CO_2/CO$ molar ratio in the hydrogenation of $CO_x$ mixtures. Although, at 470 K, theoretical conversion in pure CO hydrogenation to methanol at 75 bar is very high, >95%, in the same conditions, the allowed conversion in pure $CO_2$ hydrogenation to methanol is less than 70% [67]. In Figure 3, the calculated equilibrium compositions starting from stoichiometric mixtures (CO + 2$H_2$ and $CO_2$ + 3$H_2$) at 100 atm show a significant difference in the methanol

concentration starting from $CO_2$ hydrogenation with respect to pure CO hydrogenation and the significant amounts of water produced.

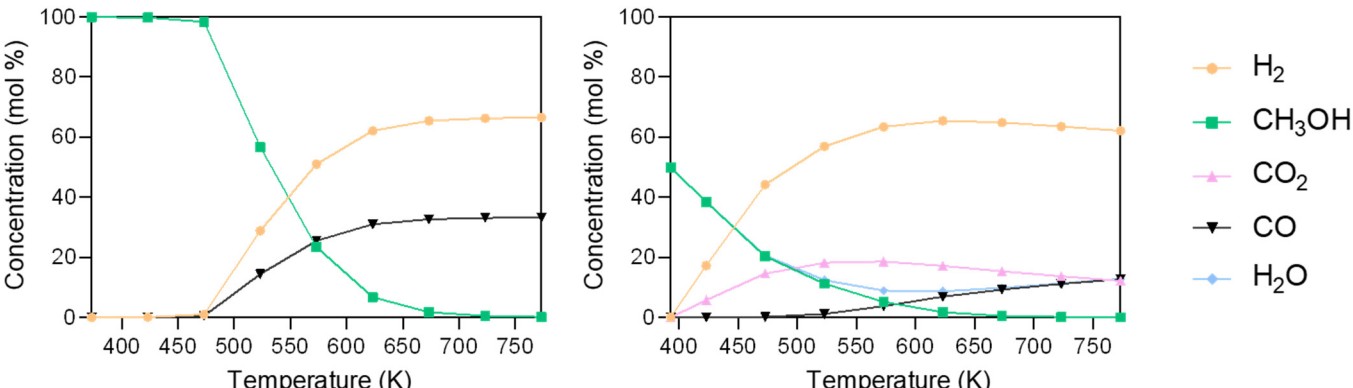

**Figure 3.** Equilibrium composition of methanol synthesis from CO (**left**) and $CO_2$ (**right**) with stoichiometric feeds, at 100 atm. Calculated using a Gibbs reactor and Soave–Redlich–Kwong equation of state and allowing a calculated phase for the reaction.

### 3.1. Conventional Methanol Synthesis

Methanol synthesis is classically realized today in Western countries by converting syngas produced from the steam reforming or autothermal reforming of natural gas [68–70]. For large-scale methanol synthesis processes, such as the Megamethanol process from AirLiquide-Lurgi [71] and SynCOR Methanol[TM] from Topsøe [72], autothermal steam reforming with a low steam-to-carbon ratio is considered best suited today. In China, the methanol synthesis gas is mainly produced by coal gasification [70,73]. The obtained syngas is essentially a mixture of CO, $CO_2$, and $H_2$, with a stoichiometry ratio, SR = $H_2$-$CO_2$/ CO + $CO_2$ in the range of 1.5–3, and a CO/$CO_2$ ratio in the range of 0.8–1.2. To adjust SR, the water-gas shift reaction (in the case of SR < 2, as typical for gases from autothermal reforming or coal gasification) or the reverse water-gas shift reaction (in the case of SR > 2 from methane steam reforming) could be realized. Instead, a high purging rate with the recovery of reactants from purge is commonly realized.

The reaction is performed at 473–523 K, 50–150 barg, using water-cooled, steam-rising, or gas-cooled reactors, sometimes more than one in series or in series and in parallel [74]. In Figure 4, the reactor system for the Megamethanol process is shown, where a multitubular gas-cooled reactor (GCR) and a multitubular water-cooled reactor (WCR) are used in series, with intermediate cooling and liquid product removal, to push thermodynamics in the GCR [71]. A main practical point concerning methanol synthesis processes is the selectivity of products. The typical byproducts of methanol synthesis from syngases are dimethylether, methylformate, higher alcohols, and hydrocarbons, with selectivities which are usually very low (of the order of hundreds of ppm) but must be minimized to reduce the complexity of the purification section.

The main suppliers all produce catalysts based on Cu/ZnO with the addition of $Al_2O_3$ and/or $Cr_2O_3$ [74,75], besides additional promoters such as Mg, Ca, Si, and Zr [35]. The catalysts are prepared by the thermal treatment of precursors obtained via coprecipitation, such as Cu-Zn mixed malachite, rosasite, or aurichalcite-type hydroxycarbonates [76], with the typical Cu/Zn atomic ratio being in the 2–3 range and minor alumina amounts [36,77]. According to Waugh [78], the best composition in terms of CuO/ZnO/$Al_2O_3$ in the unreduced catalyst is around 60:30:10. The Topsøe MK-121 catalyst was reported to contain, in the unreduced form, >55 wt% CuO, 21–25 wt% ZnO, and 8–10 wt% $Al_2O_3$ in the fresh catalyst, graphite, carbonate, and moisture balance [79]. According to Topsøe and researchers, Cu particles act as the active component in this material, while ZnO and $Al_2O_3$, more than a support, can be regarded as structural promoters that separate the Cu particles from one another, thereby diminishing particle growth. ZnO also has the additional, even more

important role of assisting Cu in producing methanol. The activity boost is related to the migration of Zn atoms onto the surface of the Cu particles: the fraction of surface atoms in the Cu particles substituted by Zn, θZn, is a function of the gas's reduction potential [80]. The development of the new FENCE™ technology allowed us to produce the more recent MK-151 FENCE™ catalyst, where the copper nanoparticles are finely dispersed in the catalyst by creating a separation between them of multiple oxide particles. Topsøe emphasizes the role of impurities in the catalyst, coming from low-purity catalyst raw materials, such as iron, chlorine, and sulfur, for high methanol selectivity. Also, Clariant (previously Süd Chemie) catalysts, denoted as MegaMax®®700 and the newest MegaMax®®800 and MegaMax®®900, are reported to be constituted of $CuO/ZnO/Al_2O_3$ [81,82].

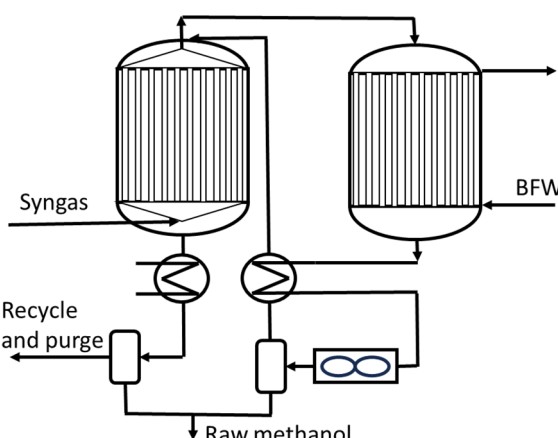

**Figure 4.** Reactor system in the Megamethanol process, see ref. [71]. BFW = boiler feed water.

A novel catalyst layering technology for enhanced methanol synthesis performance denoted as MegaZone™ Technology from Clariant and Air Liquide has been recently developed. Catalysts with moderate activity are loaded in the hotter zones of the converter to prevent hotspots, while activity-enhanced catalysts are placed further down the reaction pathway to intensify reaction rates in the lower portion of the converter. On the one hand, less thermal stress on catalysts will lead to longer catalyst lifetimes. High activity in the bottom part of the reactor will increase reaction rates and reduce by-product formation by up to 10% [81].

The composition of the Katalko 51 series from Johnson Matthey is 64% CuO, 24% ZnO, 10% $Al_2O_3$, and 2% MgO. Magnesium oxide (MgO) is incorporated to maximize the initial dispersion of copper and, therefore, boost the initial activity of the methanol synthesis catalysts. The increase in the copper surface area in a Mg-containing copper-based catalyst can be of the order of 20% versus a Mg-free formulation [83]. The addition of MgO allows for increased selectivity to methanol, reducing the coproduction of the following byproducts: methylformate, ethanol, n-propanol, 2-propanol, and 2-butanol. The addition of Si, in the form of silica, to the formulation improved long-term stability and further reduced ethanol selectivity in the new Katalko 51–102 commercial catalyst [84].

Over commercial catalysts, the maximum reaction rate is obtained with a $CO_2/CO + CO_2$ ratio of around 10%, quite typical of gases from coal gasification, while pure $CO_2$ hydrogenation is notably slower than the conversion of typical syngases but significantly faster than the hydrogenation of pure CO to methanol [67,85]. To reduce the emissions of $CO_2$ from methanol synthesis starting from natural gas, BASF patented a process that, in principle, does not emit any $CO_2$, being based on syngas produced by partial oxidation (an exothermic reaction which does not need heated furnaces), and also using $CO_2$ captured from the waste purge gases [86].

### 3.2. Methanol Synthesis from $CO_2$ through Previous Reverse Water-Gas Shift

As already said, methanol synthesis can be obtained using $CO_2$ feeds with a conventional methanol synthesis process through a previous reverse water-gas shift step [64]. However, most studies suggest that this way is not needed for methanol synthesis from $CO_2$.

### 3.3. Direct Methanol Synthesis from $CO_2$

Catalysts for conventional methanol synthesis from syngases based on the $Cu/ZnO/Al_2O_3$ system are also active for the hydrogenation of pure $CO_2$ to methanol [87]. The methanol synthesis rate on typical Cu-based methanol synthesis catalysts increases significantly by adding $CO_2$ to pure CO by up to 10% ($nCO_2/nCO + nCO_2$); then, it slows down again [67]. Thus, the reaction rate is fastest for syngas compositions (in terms of the $CO_2/CO$ ratio) similar to that of coal gasification, while it decreases for syngases with higher $CO_2/CO$ ratios than those from methane steam reforming and is even slower for pure $CO_2$ feed. In any case, methanol synthesis from $CO_2 + H_2$ is faster than methanol synthesis from $CO + H_2$.

Additionally, as obvious, methanol synthesis from $CO_x$ produces more and more water by increasing the $CO_2/CO$ molar feed ratio, which results in faster copper sintering [35]; thus, stronger hydrothermal stability is needed for the catalyst. On the other hand, the water-gas shift equilibrium (Reaction (3)) is also established in typical reaction conditions, leading to an equilibrium between CO and $CO_2$ dependent on the water-to-hydrogen ratio.

As stated, the byproducts of conventional methanol synthesis are dimethylether, methylformate, and higher alcohols and hydrocarbons. In the case of pure $CO_2$ feed, CO can also form as a byproduct through the reverse water-gas shift equilibrium. Higher methanol yields would be obtained with catalysts that catalyze methanol synthesis without catalyzing rWGS under reaction conditions [35]. However, taking into account that CO may be recycled to the reactor together with unreacted $CO_2$ and hydrogen and that $CO+CO_2$ mixtures react faster than $CO_2$, CO coproduction can be seen as a good catalyst feature-accelerating process reaction rate.

In any case, conventional industrial methanol synthesis catalysts, such as the Katalko 51 series from Johnson Matthey, also work as excellent catalysts for $CO_2$ hydrogenation to methanol. The addition of silicon in the composition of Katalko 51 series Johnson Matthey catalysts also improved its isothermal stability for application in the $CO_2$ hydrogenation methanol process [84]. Similarly, Topsøe developed a new catalyst also based on the $Cu/ZnO/Al_2O_3$ system, MK-317 SUSTAIN™, which has excellent activity, selectivity, and stability for direct methanol synthesis from $CO_2$ [88].

## 4. The Methanation Processes and the Catalysts

The relevant reactions in methanation processes are the following:

$$CO + 3H_2 \rightarrow CH_4 + H_2O \quad \Delta H°_{298} = -206.63 \text{ kJ/mol} \quad \Delta G°_{298} = -141.8 \text{ kJ/mol} \quad (4)$$

$$CO_2 + 4H_2 \rightarrow CH_4 + 2H_2O \quad \Delta H°_{298} = -164.9 \text{ kJ/mol} \quad \Delta G°_{298} = -113.2 \text{ kJ/mol} \quad (5)$$

However, they may also involve the WGS equilibrium. According to the strong exothermicity of both reactions, they are strongly favored at low temperatures already at atmospheric pressure and even more at higher pressures [89,90]. They are the reverse of steam methane-reforming reactions that become favored above ~870 K when their $\Delta G°$ becomes positive. CO methanation is more favored thermodynamically than $CO_2$ methanation at low temperatures, up to the temperature where their $\Delta G$ is negative [91]. Even at atmospheric pressure, the complete conversion of CO is possible until 673 K. At stoichiometric conditions and atmospheric pressure, the allowed $CO_2$ conversion at 673 K is of the order of 80% mol [89] and increases by decreasing temperature. Methane selectivity can be slightly decreased by some coproduction of higher hydrocarbons, of which the amount would increase with pressure and temperature [92]. Starting from

CO and $H_2$, $CO_2$, and carbon can also be produced by the Boudouard equilibrium, CO dissociation, and water-gas shift [93], while starting from $CO_2$ and $H_2$, rWGS can result in the production of CO. The methanation reactions are, in principle, also competitive with methanol synthesis but are thermodynamically much more favored at low temperatures. Carbon deposits coming from CO dissociation and Boudouard equilibrium can result in catalyst deactivation.

Two types of methanation processes are currently realized: the low-temperature process to remove residual $CO_x$ from pure hydrogen and ammonia synthesis gas and the high-temperature process to produce substitute natural gas.

### 4.1. Low-Temperature Methanation

Low-temperature methanation is usually realized in processes for preparing hydrogen for hydrogenation reactions, or $H_2/N_2$ mixtures for ammonia synthesis [3,13,94]. They are realized to reduce the concentration of $CO_x$ in such gases as much as possible because of the poisoning effect of $CO_x$ on hydrogenation catalysts. This reaction is realized in highly diluted $CO_x$ in $H_2$ mixtures ($CO_x$ concentration, most commonly < 1% mol), with CO being the predominant $CO_x$ compound in the feed, allowing the $CO_x$ concentration to be reduced to less than 5ppm in the process of gas leaving the methanator. As a consequence of the small amount of reacting molecules, the evolved heat is low, and adiabatic reactors can be used. The reaction temperature is typically 440–620 K.

Commercial catalysts for methanation in CO-rich syngases are either based on supported ruthenium or based on supported nickel [7,95,96]. A typical Ru-based catalyst is METH 150 Clariant (Süd Chemie) based on 0.3% $Ru/Al_2O_3$ [25] for very low-temperature applications (down to 440 K). Ruthenium-based catalysts are most active at low temperatures, do not require pre-reduction, and are not pyrophoric. However, they have found only a limited area of application due to their high costs.

Due to their sufficient activity, stability, and relatively low cost, Ni-based catalysts are predominantly used for conventional low-temperature methanation [3,95,96]. Typical nickel loadings are in the range of 20–45%wt as NiO, over $\gamma$-$Al_2O_3$ or metal aluminate supports, or $Cr_2O_3$ [95]. Clariant produces catalysts with 43% and 25% NiO on support (likely alumina) METH 135 and METH 134 in their oxidized forms [15]. Topsøe produces the pre-reduced PK-7R catalyst, constituted of >23% Ni wt [96] on an alumina carrier, which ensures that CO and $CO_2$ are fully converted to methane at an operating temperature of 460 K [97]. Johnson Matthey sells catalyst precursors (Katalko 11 series) constituted by 35 wt% NiO supported by refractory oxides, such as alumina–silica–lime–magnesia, strengthened with calcium aluminate cement [98]. Nickel catalysts are pyrophoric and cannot be used below 460 K to avoid the formation of $Ni(CO)_4$.

Several studies showed that the methanation of $CO_2$ is faster and more selective to methane than CO methanation, where ethane may form in small amounts over both nickel and ruthenium catalysts [99]. However, in the CO + $CO_2$ mixture, CO strongly inhibits the methanation of $CO_2$ over both Ni catalysts [100–102]. In fact, selective (or preferential) CO hydrogenation in the presence of $CO_2$ usually occurs, allowing the possibility of realizing CO abatement in syngases with an alternative process to WGS, e.g., with $Ru/Al_2O_3$ catalysts [103].

### 4.2. High-Temperature Methanation for Substitute Natural Gas Synthesis

High-temperature methanation is the conversion of a CO + $CO_2$-rich syngas, from coal gasification or biomass conversion, to produce a methane-rich gas to be used as a substitute for natural gas (SNG [66,104]). Due to the much higher $CO_x$ concentration in hydrogen (usually in the 25–35% mol range) and their high conversion degree, heat significantly evolved. A syngas of $H_2/CO = 3$ at 30 bar would result in an adiabatic temperature increase to 1196 K with an inlet temperature of 573 K [105]. To manage the first reactor, the partial recycling of the cooled product gas is realized, thus reducing CO concentration and, in parallel, the temperature increase to what can be handled by

a methanation catalyst. Also, very efficient cooling in the reactor system is carried out, and still, the reaction occurs in a high temperature range of up to 973 K. Several different processes have been proposed [104,106]. Most of them use multiple adiabatic fixed-bed reactors with interbed cooling [107]. Real industrial applications of this technology have been very few in the previous century [68,104], but several plants appear to have been recently put in operation or are under construction now, particularly in China [108]. Only Ni-based catalysts are apparently used, taking into account that the extent of the reaction under industrial conditions is more limited in heat and mass transfer rather than kinetics. With these catalysts, selectivity to methane in the full process is very high, although carbon deposition and the formation of higher hydrocarbons occur to a limited extent.

The Topsøe TREMP$^{TM}$ technology [105,109] allows producing SNG from coal gasification syngas using a series of adiabatic fixed-bed reactors with interbed cooling and recycling (Figure 5), with temperatures ranging from 523 to 973 K and pressures of up to 30 bar. T. A previous gas-conditioning step (i.e., water-gas shift) step is also realized to reduce the CO concentration. Up to 85% of the released heat from the methanation reactions is recovered as a high-pressure superheated steam. The product gas contains 94–98% methane, with a low percentage of residual hydrogen and $CO_2$ and CO < 100 ppm. The TREMP process is also part of the RNG technology allowing the production of renewable natural gas starting from biomass via biomass gasification [110].

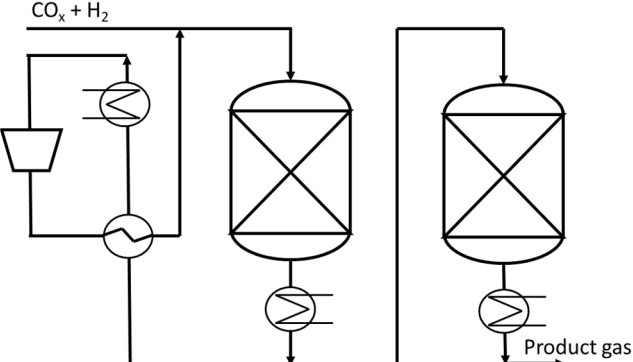

**Figure 5.** Schematics of reactor system in TREMP methanation process, see refs. [105,109].

In Topsøe's TREMP™ process, the catalyst (MCR2X) is a supported nickel catalyst containing 22 wt% Ni on a stabilized support, possibly based on $MgAl_2O_4$ [111], with a surface decreasing from 50 m$^2$/g (fresh) to 30 m$^2$/g (used) [103] and stable activity of up to 970 K [112]. Other nickel-based catalysts, MCR-8 and PK-7R, are used in medium- and low-temperature steps. Catalysts for this process need to be sufficiently active at low temperatures, resistant against sintering at high temperatures as well as to other phenomena producing deactivation, i.e., gum and $Ni(CO)_4$ formation [113].

The reactor system of the competitor process DAVY [114] from Johnson Matthey consists of two adiabatic bulk methanators, recycled to the first reactor, and in a variable number of adiabatic trim methanators in series at the end. The Johnson Matthey CRG catalysts used in the process are apparently based on Ni and refractory supports such as alumina–lanthana–magnesia [115]. They catalyze the methanation reaction and, simultaneously, the WGS reaction, thanks to the water produced by the methanation reaction.

Also, in the case of the VESTA process, several fixed-bed reactors in series are used, with a previous HTWGS step [116,117]. The catalysts, provided by Clariant [118], with a wide operating temperature range (500–973 K), are based on nickel [116]. Also, this process can be fed by syngases produced both by coal and by biomass gasification.

### 4.3. $CO_2$ Methanation

The methanation of captured $CO_2$ using green hydrogen is considered a possible method to valorize and reuse it, which is also in view of the so-called "power to Gas"

technologies [119,120]. Many projects and plants of various sizes have been undertaken in the last fifteen years [121,122]. The new processes under development today for $CO_2$ methanation [105,123,124] are strictly related to the ones described above for SNG production from CO-rich syngases. For example, the VESTA process is also applied as the methanation step of power-to-gas plants [117].

According to its large exothermicity, the theoretical thermodynamic methane yield approaches totality at temperatures of up to 473 K at moderate pressures [89,90,125]. At higher temperatures, $CO_2$ conversions and methane yields from thermodynamics decrease. However, methane synthesis is competitive with the production of CO through rWGS (Reaction –(1)), of which its equilibrium is shifted to $CO + H_2O$ more when the temperature is higher. Studies show high selectivities of pure $CO_2$ methanation, with a possible coproduction of small amounts of CO due to the rWGS reaction. In the presence of CO, the conversion of $CO_2$ is inhibited, but the conversion of CO to methane occurs [99,126]. Reactor modeling studies suggest that, despite the high temperature, at moderate pressures, the amount of CO will always be very low in $CO_2$ hydrogenation reactors [111,127].

Although an enormous number of projects are under development [122], information on real industrial applications is scarce. One of the earliest examples of power-to-gas technology was Audi's e-gas plant in Werlte, Germany, which was constructed by Etogas, to produce renewable synthetic methane from hydrogen and carbon dioxide emissions to achieve low-carbon mobility for Audi's A3 Sportback g-tron vehicles. Clariant provides the catalyst to be applied in such an application [128]. No detailed information is available on the Clariant's SNG 100 DCARB methanation catalyst just developed for power-to-gas processes: it seems that it is based on nickel [116]. In fact, taking into account that practical reaction temperatures in the catalyst bed are relatively high, the low-temperature activity of noble metals such as ruthenium is not necessary, at least in main methanation reactors. Thus, Ni-based catalysts are most promising [129]. High-temperature stability is certainly a key feature. A very large number of different catalyst compositions have been proposed in the literature for $CO_2$ methanation. A critical analysis of the scientific literature data concerning catalysts for $CO_2$ methanation allowed us to propose that $Ni/Al_2O_3$ catalysts with medium metal loading allowing small metal particles to almost entirely cover the support surface, with additional ceria and/or basic oxide dopants as the stabilizers and activating components, would be very active in methanation and not produce many carbon residues and gums [130].

## 5. The Production of Higher Hydrocarbons

The most relevant reactions in the production of higher hydrocarbons from $CO_x$ hydrogenation, usually denoted as the Fischer–Tropsh processes, are the following:

$$CO + 2H_2 \rightarrow -CH_{2-} + H_2O \qquad \Delta H^{\circ}_{298} = -165 \text{ kJ/mol} \qquad (6)$$

$$CO_2 + 3H_2 \rightarrow -CH_{2-} + 2H_2O \qquad \Delta H^{\circ}_{298} = -125 \text{ kJ/mol} \qquad (7)$$

$$2CO + H_2 \rightarrow -CH_{2-} + CO_2 \qquad \Delta H^{\circ}_{298} = -204 \text{ kJ/mol} \qquad (8)$$

which are strongly exothermic equilibrium reactions favored at low temperatures and high pressures. Reactions (6) and (7) are the reverse of the steam reforming of higher hydrocarbons, while Reaction (8) is the reverse of the dry reforming of higher hydrocarbons. In the low-temperature process with used catalysts, cobalt-based linear paraffins and the corresponding linear olefins are the products. According to thermodynamic analyses [131,132], product distribution is very sensitive to feed composition, and to temperature and pressure to a lesser extent. An increase in the $H_2/CO$ feed ratio and temperature results in a shift in selectivity toward low carbon number hydrocarbons, especially towards methane. Conversely, an increase in pressure favors high carbon number hydrocarbons. In fact, the high-temperature process is realized to produce gasoline-range hydrocarbons, olefins, and some oxygenated compounds.

### 5.1. Conventional Low-Temperature Fischer-Tropsch (LTFT) Processes

The LTFT process, performed at 473–523 K, 20–30 bar, allows us to convert syngases into middle distillate and hydrocarbon waxes, both applicable to produce a high cetane number, sulfur-free Diesel fuel, and jet fuel [133]. Cobalt-based catalysts are used in most LTFT industrial technologies [134–136]. In fact, among the different metals active as FT catalysts (essentially ruthenium, iron, cobalt, and, to a lower extent, nickel), cobalt represents the best choice for this application for its moderate price, good selectivity to high molecular weight paraffins, and low selectivity to methane, olefins, and oxygenates [137]. According to IFPEN researchers, among possible supports, alumina is a good choice because of its availability with texture suitable to slurry conditions in waxy liquids. The cobalt FT catalysts are usually supported on high surface area alumina (150–200 $m^2$/g) and typically contain $15 \div 30$ wt% of cobalt. This means that cobalt is largely dispersed on the support surface forming, in the reduced state, very small cobalt metal particles [138,139]. To stabilize them and decrease selectivity to methane, these catalysts also contain small amounts of noble metal promoters (typically 0.05–0.1% weight of Ru, Rh, Pt, or Pd) and/or an oxide dopant as well (potash, zirconia, lanthana, cerium oxide, 1–10%) [140]. The role of noble metal doping has been supposed to be to improve the reducibility of cobalt, while the support and the oxide promoters may have a role in controlling the size of the cobalt metal to an optimal value. The slurry phase distillate (SPD) process from Sasol uses a Co-based Sasol proprietary catalyst typically operating at 500 K and 2.5 MPa [133], apparently supported by alumina [141]. Upon working, part of the catalyst is converted into the carbide $CoC_2$, with moderate deactivation [141] but increased olefin production [142,143].

In a recent study, researchers from Johnson Matthey [144] showed that different supports for cobalt may significantly influence the stability of the catalyst and selectivity to products, with the titania- and zirconia-supported catalysts producing predominantly paraffins, whereas $CeO_{2-}$ and $ZnO^-$ containing catalysts produce significantly more olefins and alcohols, showing that the product distribution may be tailored depending on the support material. The catalysts used by Shell in their GTL technology were reported to be likely constituted by cobalt on $ZrO_2$-$SiO_2$ support [145]. With cobalt catalysts, the optimum ratio of the reactants $CO + CO_2/H_2$ is 1:2, that is, the stoichiometric ratio for FT. $CO_2$ as well is involved in the reaction.

The ability to catalyze hydrocarbon chain growth is a key feature of Fischer-Tropsch catalysts. Although the molecular mechanism for chain growth on cobalt catalysts is still far from ascertained, data indicate that the high pressure of CO favors it and that adsorbed CO is certainly a promoter of chain growth [146].

Water-cooled multi-tubular fixed-bed reactors (FBR) or slurry bubble column reactors (SBCR) are used industrially [135,147]. Recently, Johnson Matthey developed a new reactor technology for its FT CANS process, based on fixed-bed reactors, termed CANS$^{TM'}$, consisting of modular catalyst containers providing modified reactant flow paths [144].

Iron catalysts can also be applied at low temperatures in an LTFT process, as carried out years ago by Sasol [145]. Due to their high activity in the water-gas shift reaction, iron catalysts are suited for FT syntheses using syngases characterized by a quite low $H_2/CO_x$ ratio (~1.7 mol), as those arising from coal gasification. They produce more olefins and oxygenates than cobalt catalysts. Thus, Co-based catalysts appear to be the most performant, particularly for the synthesis of Diesel-type fuels, and have higher $H_2/CO$ ratio syngases (e.g., from natural gas steam reforming). However, Co-based catalysts are less resistant to some poisons, such as ammonia [148], which can be present, e.g., in syngases from biomass gasification.

### 5.2. Conventional High-Temperature Fischer-Tropsch (HTFT) and Medium-Temperature Fischer-Tropsch (MTFT) Processes

The high-temperature Fischer-Tropsch process (HTFT) [149] consists of the conversion of syngases at 590–630 K and 20–40 bar, aimed at the production of low molecular mass alkenes and liquid products primarily in the gasoline range, with the coproduction of

several oxygenate compounds. Quite recently, an MTFT process has been developed in China, working at 530-570 K [150,151]. Both HTFT and MTFT are performed on iron-based catalysts [149,150,152,153]. These processes produce more oxygenates and olefins and a slightly different molecular weight distribution for hydrocarbons compared to Co-based catalysts applied in the LTFTS [149,151].

A typical iron FT catalyst is unsupported but also contains a low percentage of silica, copper, and potassium. Copper is added to aid in the reduction of iron, while silica is a structural promoter added to stabilize the surface area but may also have a chemical effect on the catalyst properties. Potassium is considered to increase the catalytic activity for FT synthesis and water-gas shift reactions, promote CO dissociation, and enhance chain growth, increasing olefin yield and lowering the $CH_4$ fraction [154,155]. Under reaction conditions, the catalyst converts into mixtures of carbides like $\chi$-$Fe_5C_2$, $\varepsilon$-$Fe_2C$, $\theta$-$Fe_3C$ [156,157], and magnetite $Fe_3O_4$, with only small amounts of $\alpha$-Fe. Studies using isotopically labeled $^{13}CO$ as a reactant or, the reverse, $^{13}C$-labeled catalyst and $^{12}CO$ as a reactant indicate that the carbon atoms of hydrocarbon products come from the reactant CO, while surface carbon atoms of the carbide may participate in the initiation step of chain growth [158]. In any case, carbidic rather than metallic catalysis should really occur in the case of HTFT synthesis. Also, in the case of iron carbide catalysis, it seems clear that CO is a promoter of chain growth, with a possible role of a CO insertion step in the chain growth mechanism [159].

*5.3. Fischer-Tropsch Processes Using Electrolytic Hydrogen and Captured $CO_2$*

The Fischer-Tropsch processes can be of interest to convert pure $CO_2$ feed with electrolytic hydrogen to e-fuels, such as e-jet fuel and e-Diesel fuel. This can be accomplished in two ways [160]: (i) a direct way, i.e., feeding directly the $CO_2$ + $H_2$ mixture to the catalyst (the so-called $CO_2$-FTS); (ii) an indirect way, i.e., sequencing the rWGS reaction step producing a CO + $CO_2$ containing syngas with a following conventional FT step, either LTFT or HTFT.

The direct $CO_2$-FTS reaction is slower than CO-FTS and conventional FTS reactions under the same conditions and tends to produce smaller chain hydrocarbons. This is likely related to the main role of carbon monoxide in dissociation-producing active carbide species as well as in the promoting role of CO in the chain growth steps. This, however, influences far more cobalt- than iron-based catalysts. At 490 K and 20 bar, iron catalysts produce long-chain olefins, while cobalt-based catalysts act mostly as methanation catalysts [161]. The different behavior is explained by taking into account that iron catalysts have high WGS/rWGS activity, thus allowing the production of the real reacting and chain growth promoter molecule CO from $CO_2$ in FTS conditions. It has been reported that catalysts based on delafossite, $CuFeO_2$, produce copper-doped iron-carbide-based catalysts for this process, giving rise to high selectivities to hydrocarbons in the Diesel fuel and wax range [162]. When combined with the HZSM5 zeolite, this catalytic material allows the production of gasoline-rich [163] and aromatic-rich [164] hydrocarbon mixtures. Using a Na-containing Co-Fe catalyst, high selectivity to kerosene/jet fuel has been obtained [165]. Apparently, only laboratory tests are available today for this reaction.

On the other hand, the indirect method, through the previous rWGS reaction step, followed by a more or less conventional LTFT step, seems to be the industrial choice today. In fact, several companies, such as Topsøe [56], Johnson Matthey, London, UK [61], Axens, Rueil-Malmaison, France [63], and Clariant with Ineratec [60] and Audi, Zwickau, Germany [166], are developing processes based on this scheme, with a previous high-temperature rWGS step (as mentioned above) and a nearly conventional LTFT step later.

## 6. Mechanistic Aspects of $CO_2$ Hydrogenations

The mechanisms involved in $CO_x$ hydrogenation imply the role of metallic phases, as well as of oxide and carbide phases. To have an idea of the role of these phases, a look at the surface chemistry and physics of the different catalyst components is needed. Concerning

oxide phases, discussion can be based on their ionicity and acido-basicity [167,168], as well as their reducibility [169]. On the other hand, the catalytic behavior of transition metals in reaction conditions is strongly related to the position of the d-band center, in agreement with the d-band model of Hammer and Nørskov relating adsorption energies to the d-band position, and the adsorption energies to barriers in catalytic reactions [170–172]. In Table 2, data concerning the interaction of the most relevant catalytic metals with the reactants of the processes discussed here are summarized.

**Table 2.** Surface physical chemistry data concerning the interaction of catalytic transition metals with carbon oxides and hydrogen, and the d-band center value.

| | | d-Band Center | Hydrogen Adsorption | | Strongest CO Adsorption | | | Fastest CO Dissociation | | | Strongest $CO_2$ Adsorption | | | Price |
|---|---|---|---|---|---|---|---|---|---|---|---|---|---|---|
| | | | $\Delta E_{FCC}$ | $\Delta E_{ontop}$ | $E_{ads}$ | Face | Geometry | $E_{ads}$ | $E_{att}$ | Face | $E_{ads}$ | Face | Site | |
| | | eV | eV | eV | eV | | | | | | | | | USD/lb |
| Cu | Fcc | −2.67 | 0.07 | 0.62 | −0.68 | 011 | short bridge | 1.81 | 2.68 | 011 | 0.25 | 332 | terrace | 3.67 |
| Ni | Fcc | −1.29 | −0.37 | 0.19 | −1.84 | 111 | threefold | −0.07 | 1.64 | 001 | −0.38 | 332 | terrace | 7.79 |
| Co | Hcp | −1.17 | −0.31 | 0.28 | −1.58 | 11–20 | short bridge | 0.40 | 1.33 | 11–20 | −0.59 | 015 | terrace | 15.16 |
| Fe | Bcc | −0.92 | −0.54 | 0.23 | −2.20 | 011 | threefold | −1.38 | 1.08 | 011 | −1.23 | 321 | step | <<1 |
| Pd | Fcc | −1.83 | −0.42 | 0.11 | −1.77 | 111 | threefold | 1.40 | 2.83 | 001 | −0.10 | 332 | terrace | 16,289.28 |
| Ru | Hcp | −1.41 | −0.41 | −0.10 | −1.95 | 10–10 | terminal | 0.18 | 1.38 | 11–20 | −0.90 | 015 | step | 7440.00 |
| Pt | Fcc | −2.25 | −0.37 | −0.38 | −1.81 | 001 | bridge | 2.27 | 4.09 | 111 | −0.07 | 332 | terrace | 14,206.72 |
| Ref. | | [173] | [174] | [174] | [175] | [175] | [175] | [175] | [175] | [175] | [176] | [176] | [176] | [177] |

Another relevant point in developing industrial catalysts is their relative cost. In particular, the use of expensive metals, like platinum group and noble metals, even when they can be used in much lower amounts than cheaper transition metals, due to their great price differences, can be inconvenient when performance advantages are limited. For this reason, today's price of metals is also added to Table 2.

*6.1. The Adsorption and Activation of Hydrogen*

A number of oxides that can be used as supports or components of hydrogenation catalysts significantly adsorb hydrogen and show some useful activity as catalysts for hydrogenation and dehydrogenation reactions. This is the case of ZnO, $Fe_3O_4$, $Cr_2O_3$, $Ga_2O_3$, $ZrO_2$, and $CeO_2$ [178]. Besides very weak and reversible molecular adsorption occurring only at very low temperatures, at moderate temperatures, the adsorption of hydrogen on several metal oxides was reported to be heterolytic, occurring on exposed cation–oxide couples. Thus, hydrogen dissociation produces a metal hydride species on cationic centers and a new hydroxyl group on an oxide site. The best-known case is that of ZnO, which has been the subject of a number of studies [179–181]. This oxide is an active catalyst in hydrogenations, such as methanol synthesis [182], as well as in alcohol dehydrogenation reactions [178]. Well-evident surface hydride species have been observed to be formed by $H_2$ adsorption at room temperature by vibrational spectroscopies (IR, HREELS, INS), both of the terminal Zn-H type I (1700–1600 cm$^{-1}$), which is more weakly adsorbed, and of the bridging Zn-H-Zn type II (1450–1200 cm$^{-1}$), which is more strongly adsorbed, formed together with new OH groups. Spectroscopic studies confirmed that some of the hydride species are indeed active in hydrogenation, e.g., producing CHO formyl species from CO as a possible intermediate in methanol synthesis over ZnO [183]. The heterolytic dissociation of hydrogen-forming active metal hydride species has also been observed on chromia ($Cr_2O_3$), a component of hydrodealkylation and paraffin dehydrogenation catalysts [178], where only terminal hydride species were found ($\nu$Cr-H 1714, 1697 cm$^{-1}$ [184]) and the combination of ZnO and $Cr_2O_3$, i.e., zinc chromite ($ZnCr_2O_4$) catalysts [184]. The latter is the active phase of the old BASF industrial catalysts for methanol synthesis with the high temperature and pressure process, operated at above 300 bar and 573–673 K [70], industrially used until the seventies of the last century. On $Cr_2O_3$ and

on other mixed chromites such as Co-Cr and Mn-Cr oxides, both terminal and bridging species were found [184]. Two terminal hydride species ($\nu$Ga-H 2003 and 1980 cm$^{-1}$) have also been found to be formed by hydrogen adsorption on gallium oxide polymorphs [185] and assigned to hydride species bonded to tetrahedral and octahedral Ga ions, respectively. Dissociative adsorption of hydrogen was also observed on zirconia: to terminal hydride species was assigned a sharp $\nu$Zr-H band at 1562 cm$^{-1}$, while a band at 1540 cm$^{-1}$ was assigned to a dihydride H-Zr-H species, and a broad band at 1371 cm$^{-1}$ was assigned to the bridging species. (ZrHZr) [186–188]. The dissociative adsorption of H$_2$ was also observed to occur in small amounts on the defect sites of MgO. Calculations allowed us to assign a band of reversible species at 1325 cm$^{-1}$ to the terminal Mg-H hydride group on tri-coordinated magnesium. The hydrogen atoms bridging two or three neighboring, low-coordinated Mg atoms of the surface are suggested to be responsible for complex bands in the 1130–880 cm$^{-1}$ region [189].

A different mechanism has been reported to occur over more easily reducible oxides such as CeO$_2$, or at higher temperatures, i.e., the reductive homolytic adsorption of hydrogen-producing hydroxy groups and near-surface reduced metal centers [190,191]. In this case, an "inverted" version of the so-called Mars–van Krevelen mechanism, or redox mechanism, can occur in hydrogenation catalysis [192], such as in hydrodeoxygenation reactions. The surface, which is reduced by hydrogenation, may be re-oxygenated by the substrate, which is consequently de-oxygenated. DFT calculations have shown that H$_2$ may adsorb through homolytic dissociation on CeO$_2$(111) with a relatively low activation barrier (0.2 eV) and strong exothermicity [193]. More recent studies showed that hydride species may form by hydrogen adsorption on reduced ceria CeO$_{2-x}$, thus concluding that both heterolytic and homolytic H$_2$ adsorption can occur on ceria [194]. Theoretical studies indicate that a similar conclusion can be obtained for H$_2$/TiO$_2$ systems (anatase and rutile) [195] and for the H$_2$/Fe$_3$O$_4$ system [196], i.e., that both heterolytic and homolytic dissociation can occur and that the heterolytic product and the hydride M-H species can play a role in the homolytic dissociation mechanism. However, experimental data are lacking for these systems. In more recent years, the hydrogenation activity of indium oxide In$_2$O$_3$ has also been established. Also, in this oxide, heterolytic dissociation is supposed to occur, although the resulting species seem to be IR and Raman inactive [197,198]. It is possible that also oxides that adsorb heterolytically hydrogen at low temperatures, undergo reductive adsorption at higher temperatures, heterolytic dissociation being the first step in homolytic dissociation mechanism. On the other hand, several transition metal oxides, can be reduced to lower oxides and to metals by hydrogen at moderately high temperatures [199], thus they are essentially not present in practical hydrogenation catalysis conditions. In practice, iron, cobalt, nickel, and copper oxides do not exist as stable phases in most catalytic hydrogenation conditions, while cerium, zinc, zirconium, and chromium oxides may exist as slightly reduced phases. Typical nonreducible oxides, thus being stable even in significant reducing conditions, are alumina, silica, magnesia, etc.

As for most common insulating supports, the adsorption of hydrogen on silicas is negligible at room or higher temperatures [200] in normal conditions. The adsorption and reactivity of hydrogen on alumina have been the subject of early studies [201–203], showing the formation of different species. Adsorbed species stable at high temperatures are observed. In particular, Kramer and Andre [203] showed that atomic hydrogen can be spilled over from metallic centers to alumina and remain stable until 750 K. More recent theoretical and experimental studies suggest that spilled-over hydrogen can be adsorbed over $\gamma$-Al$_2$O$_3$-forming hydride species likely bonded on trigonal Al$^{3+}$ centers [204–206]. It must, however, be taken into account that these studies fully neglect the possible role of transition metal impurities unavoidably present on and in $\gamma$-Al$_2$O$_3$, such as, in particular, the non-negligible amount of Fe$^{3+}$ and Cr$^{3+}$ [207], that can either be reduced by hydrogen or involved in H$_2$ dissociation. On the other hand, hydrogen spillover is far more evident on reducible oxides than on alumina [208].

Quantitatively, the adsorption of hydrogen in non-reducible oxides ($\gamma$-Al$_2$O$_3$) and weakly reducible oxides (TiO$_2$ and ZrO$_2$) appears to be essentially negligible at temperatures up to 873 K, while it is considerable on reducible oxides such as on CeO$_2$ above 573 K and on Fe$_2$O$_3$ above 673 K [209]. In the latter case, bulk reduction to Fe$_3$O$_4$, FeO, and metallic Fe also occurs depending on temperature and hydrogen pressure.

On the other hand, hydrogen activation over metal/oxide catalysts is mostly considered to occur on metallic centers [210], and when supports are active in adsorbing it. For example, the amount of hydrogen adsorbed on a ZnO-ZrO$_2$ support is negligible with respect to the hydrogen adsorbed on copper [211]. A similar phenomenon occurs for Pt on several oxides [209]. In fact, hydrogen dissociatively adsorbs very quickly on almost all metal surfaces, with the occupation of on-top, bridge, or hollow sites by atomic hydrogen species [212]. On most metal surfaces, the dissociation of hydrogen is only weakly activated or even barrierless. For example, it has been found that when a H$_2$ molecule chemisorbs on a Pt surface, the antibonding $\sigma^*$ orbital of H$_2$ is completely filled by electrons from platinum, leading to its homolytic dissociation, which is not kinetically hindered [213]. Most hydrogen molecules are expected to stay on the surface because of the relative instability of subsurface hydrogen compared to surface hydrogen on most metals, except Pd [214]. In all cases [215], hydrogen dissociation gives rise to strongly bonded surface atomic hydrogen, mostly occupying hollow sites. Studies mostly devoted to (111) surfaces of face-centered cubic metals, (0001) surfaces of hexagonal close-packed metals, and (110) faces of body-centered cubic metals show the resulting predominant occupancy 3-fold sites, although other adsorption sites, such as bridge sites and top sites, may be competitive with 3-fold sites [216]. Only in the case of Ir(111), top sites appear more favored than hollow sites for surface atomic hydrogen location.

On the other hand, it must be considered that on group 11 metals (Cu, Ag, Au), hydrogen dissociation is significantly activated and endothermic [174]. Adsorbed hydrogen is definitely less stable on copper than on the other metals of interest here (nickel, cobalt, iron) [174]. Hydrogen adsorption strength follows the trend of the position of their d-band (Table 2).

Indeed, the size and shape of metal particles [217] and the nature of the support have effects on the nature of adsorbed hydrogen. DFT calculations [218] show that the adsorption energy of hydrogen should increase for Pt particles on ionic (basic) supports. As for nickel powder, TPD studies [219] evidenced three distinct forms of hydrogen permanently chemisorbed at 100 K: $\gamma$-form, located in the subsurface layer and desorbed at about 186 K; $\beta$-form, adsorbed in the so-called second layer and evolved at about 327 K; and $\alpha$-form, fixed directly on nickel surface and desorbed at the 350–670 K range. Alumina and silica supports insignificantly affect hydrogen strongly adsorbed on nickel but significantly affect weakly adsorbed hydrogen [220]. On the other hand, the nature of the supports also affects the metal particle size and stability of nanoparticles, as well as their electronic properties [221].

An additional point concerns the role of the contamination and/or doping of metallic surfaces on hydrogen adsorption. One relevant example is the Cu-ZnO$_x$ case, where it seems that the ZnO$_x$ species strongly interacting with copper may favor hydrogen dissociation and act as a reservoir of atomic hydrogen [222].

As shown above, metal carbides have a role in the catalysis of Fischer-Tropsch syntheses. Relatively few data are reported on the adsorption of hydrogen on transition metal carbide phases. A DFT study of hydrogen adsorption on the low-index surfaces of four major carbides, TiC, VC, ZrC, and NbC, indicates that it is generally exothermic and occurs predominantly on the surface carbon atoms, producing surface CH bonds [223]. In fact, the stretching vibrations of hydrocarbon species were observed when an iron sample previously carbided by syngas treatment, was treated with H$_2$ flow [224]. A computational study of the interaction between hydrogen and eight primary surfaces of Fe$_5$C$_2$ nanoparticles allowed us to conclude that dissociative adsorption is favorable due to spontaneous dissociation on iron-rich surfaces, as well as with a lower barrier (<0.5 eV) on carbon-rich

surfaces. The single hydrogen atom preferred to be located at C sites on C-rich surfaces and Fe sites on iron-rich facets [225].

*6.2. The Adsorption and Activation of CO*

Carbon monoxide is a very weak base that adsorbs at low temperatures, preferentially on electron-withdrawing centers on oxide surfaces (i.e., both Lewis acidic and Brønsted acidic centers) through the lone pair at its C atom [226,227], although very weak O-bonded species can also sometimes be observed at very low temperatures [228]. Infrared spectroscopic studies of CO adsorption on oxide materials have been reported by many authors because they are very informative for the characterization of active centers on metal oxide surfaces [229,230]. On covalent oxides, such as amorphous silica, very weak reversible adsorption on surface OH groups (silanols) is observed. On ionic oxides, molecular end-on (terminal) adsorption is observed in cationic centers. In the case of $d^0$ cations, the position of the CO band is sensitive to the cation Lewis acid strength that enhances the C-O stretching frequency as a result of electron withdrawal through the σ-bond at the C-end. The CO stretching shifts from the 2140 cm$^{-1}$ position of unperturbed CO to >2200 cm$^{-1}$ for CO interacting with the strongest Lewis sites ($Al^{3+}$ ions of alumina-based materials). Such a terminal CO adsorption is observed at low to very low temperatures and is usually weak and reversible at room or slightly higher temperatures. In the case of cations with partially filled d orbitals, d - π* electron backdonation can also occur, and this tends to shift down the CO stretching frequency but may increase adsorption bond strength significantly. This is the case of, e.g., CO's interaction with cuprous ions, $Cu^+$, with CO stretching that is found at 2120–2100 cm$^{-1}$. Polycarbonyl species, e.g., dicarbonyl and tricarbonyl species, can also be observed to sometimes form over metal cations [225]. In any case, the molecular adsorption of CO on metal oxides is quite a weak phenomenon, definitely weaker than on zerovalent metal centers [231].

At moderately high temperatures, CO adsorption on hydroxylated ionic metal oxides may give rise to formate ions through reactions with hydroxy groups with "associative" adsorption mechanisms [232,233]. Only at very high temperatures, carbon deposition from CO is observed on metal oxides together with $CO_2$, attributed to the reverse Boudouard reaction [234].

Only when highly oxidizing cations are present at the surface (like $Ru^{4+}$, $Pd^{4+}$, $Cu^{2+}$, $Pt^{2+}$, and $Co^{3+}$ ions), CO oxidative adsorption also gives rise to adsorbed $CO_2$ in its different forms (see below) at very low temperatures. On the other hand, all reducible oxides, such as Fe, Co, Ni, and Cu oxides, are reduced by carbon monoxide to lower oxides and/or to the corresponding metals [235] or converted to metal carbides at relatively high temperatures, producing $CO_2$ as a co-product. In any case, as already stated, reducible oxides do not exist in strongly reducing conditions, such as in most hydrogenation reactions.

Experimental studies show that CO can adsorb on metallic centers in at least three forms: terminal (on-top), bridging, and triply (or multiply) bridging. Also, the terminal end-on adsorption of CO on on-top metallic centers [226,236] is primarily due to the σ-type interaction of the doublet at the C atom with empty orbitals of the metal ion, coupled to the backdonation from filled d orbitals of the metal center to the empty antibonding 2π* orbitals of CO. The latter results in a weakening of the CO bond and a lowering of its stretching frequency falling in the range of 2120–1850 cm$^{-1}$. The interaction of a CO molecule with multiple metal atoms is associated with an even more efficient M→C π backdonation and results in a further significant weakening of the CO bond down to the range of 1950–1600 cm$^{-1}$ [237,238]. Backdonation strength increases by decreasing nucleus charge in the transition metal rows (i.e., from right to left in the element table, following the position of the d-band; see Table 2) and also by increasing the principal quantum number of external electrons (i.e., from top to bottom in the elemental table). In fact, the interaction of CO is very weak on metallic copper, where it is dominated by the σ-donation, as the π backdonation is very weak in the case of copper and other group 1b metals, such as Ag and Au [239]. The adsorption of CO on other first-row transition metals, particularly Ni,

Co, and Fe, where bridging and triply bridging species may also be observed, is definitely stronger than on Cu [236], where only terminal on-top species form [240]. As for copper, the adsorption of CO is far stronger on $Cu^+$ than on $Cu°$ [241], being a rare case where CO adsorption is stronger on oxide than on metal.

The strong reactivity of CO with some metal surfaces can result in the possible formation and evolution of gaseous carbonyl complexes, like in the case of the production of $Ni(CO)_4$, which may form whenever Ni-based catalysts are reduced and exposed to CO at temperatures below 473 K [242]. A less easy case is the formation of iron and cobalt carbonyls. A possible intermediate state is that of highly dispersed (single) metallic atoms where surface polycarbonyl species may form by CO adsorption, such as for highly dispersed Ni on alumina [243].

Another result of this strong interaction is the occurrence of CO dissociation on the surface, producing carbon and oxygen species [175]. The CO dissociation rate is fundamentally influenced by two main factors [244]: (i) an electronic effect on the dissociation barrier, i.e., the increase in the dissociation barrier from left to right in the periodic table, which has been correlated with the metal d band center [170–172] (see Table 2); (ii) an effect of particle and surface morphology, being lower the dissociation barrier on steps, kinks, and defects than on terraces. As a common feature in catalysis, the activation energy for CO dissociation is correlated inversely with the adsorption strength, which is definitely larger for CO on platinum group metals than on group 1 elements (Cu, Ag, and Au [236]), and, for the metals of interest here, follows the trend Fe > Ni > Co >> Cu.

The formation of carbon species can result in the bulk conversion of metal into carbide. According to Chai et al. [245], Raney iron CO dissociation occurs already starting from 120 K, while partial carburization in pure CO occurs from 490 K, first producing $\chi$-$Fe_5C_2$. To obtain full carburization, $H_2$ is needed to remove oxygen coming from CO dissociation as $H_2O$. Such reactions can be favored by opportune doping: in particular, potassium doping favors CO dissociation on iron and its carburization [246]. Also, cobalt can be carburized to $CoC_2$ using pure CO at 490–500 K, while only partial carburization occurs using syngas [142,247]. Only the partial carburization of nickel to $Ni_3C$ was found with pure CO at 538 K [248].

*6.3. The Adsorption and Activation of $CO_2$*

$CO_2$ can adsorb on solid oxides in a molecular form or in a reactive form [249,250]. Molecular adsorption occurs through one of the electron doublets at an oxygen end in a linear way. This adsorption may occur on hydroxy groups through hydrogen bonding or at Lewis acid sites. This interaction is definitely weak and reversible at room or slightly higher temperatures. On ionic metal oxides, $CO_2$ may also adsorb in a reactive way, producing weakly adsorbed but strongly perturbed $CO_2$ molecules, and carbonate and bicarbonate (monohydrogencarbonate) species with different geometries and adsorption strength. Bicarbonate species are well evident on strongly hydroxylated and weakly basic oxides, such as alumina, while different types of carbonate species, i.e., monodentate, bidentate, or bridging and polydentate form on more basic surfaces. On very basic materials such as alkali earth oxides, the interaction of $CO_2$ can result in bulk carbonation even at very low temperatures. The desorption of carbonate from surfaces (including subsurface and bulk) can be taken as a measure of the basicity of the oxide [168]. Considering most common oxide carriers, $CO_2$ adsorption is practically non-occurring on silica, as on most covalent oxides (e.g., $V_2O_5$, $MoO_3$, $WO_3$), quite weak on alumina (mainly in the form of bicarbonates) and titania, and definitely stronger on zinc oxide, zirconia, ceria, magnesia, in all cases forming strongly bonded carbonates and less strongly bonded bicarbonates, while bulk carbonation easily occurs on strongly basic oxides such as CaO.

$CO_2$ adsorption on metal surfaces has been the subject of early spectroscopic studies on monocrystal surfaces [251–253]. These studies report strong spectroscopic evidence for the formation, besides physisorbed $CO_2$ only at a very low temperature, of negatively charged bent $CO_2^{\delta-}$, which, depending on the nature of the metal, may dissociate into CO

and O or transform into $CO_3^{2-}$ + CO. The presence of surface adatoms may dramatically influence the adsorption and reactivity of $CO_2$. In particular, alkali adatoms increase the binding energy of adsorbed $CO_2$ and promote the dissociation and/or the transformation of $CO_2$ into CO + O. Theoretical studies of $CO_2$ adsorption on transition metals [176,254,255] indicate that $CO_2$ exothermic adsorption strength forming bent $CO_2^{\delta-}$, the charge of this species, and the exothermic dissociation energy to produce CO-O follow the trend Cu < Ni < Co < Fe, i.e., again, the trend of d-band position (Table 2). In fact, $CO_2$ adsorption was reported to be only molecular and very weak, also in the presence of hydrogen, on flat copper surfaces [256], but dissociation is reported to occur on stepped surfaces [257]. A recent ambient-pressure X-ray photoelectron spectroscopy study [258] indicated that carbonate species, adsorbed CO, and graphitic carbon form on both Ni(111) and Ni(100) surfaces in the presence of 0.2 Torr $CO_2$. However, more than 90% of adsorption species on the Ni(111) surface are carbonate, whereas the Ni(100) surface is mainly covered by adsorbed CO and graphitic carbon.

## 7. Mechanistic Aspects and the Role of the Different Catalyst's Components

The mechanism of the WGS reactions is still controversial [12]. Regarding the HTWGS reaction over $Fe_3O_4$-based catalysts, it seems that there is a general agreement concerning the so-called redox mechanism, where carbon monoxide can first adsorb end-on on iron ions and then reduce $Fe^{3+}$ ions, while water would reoxidize them in the second step.

$$2Fe^{3+} + O^{2-} + CO \rightarrow 2Fe^{2+} + CO_2 \tag{9}$$

$$2Fe^{2+} + H_2O \rightarrow 2Fe^{3+} + H_2 + O^{2-} \tag{10}$$

where the second step is the reverse of the reductive homolytic dissociative adsorption of hydrogen on iron oxides (see above). As seen, new catalysts based on $ZnAl_2O_4$ have also been developed for HTWGS to be used at low steam-to-carbon ratios. In this case, the so-called associative mechanism, through formate ions, seems to be most likely:

$$Zn^{2+} + {}^-OH + CO \rightarrow Zn^{2+} + HCOO^- \rightarrow Zn^{2+}\,{}^-H + CO_2 \tag{11}$$

$$Zn^{2+}\,{}^-H + H_2O \rightarrow Zn^{2+} + {}^-OH + H_2 \tag{12}$$

where, again, the second step is the reverse of the heterolytic dissociation of hydrogen on ZnO (see above) [259]. In both cases, oxide catalysis certainly occurs.

More controversy exists concerning the mechanism of LTWGS on Cu-ZnO-$Al_2O_3$ catalysts, which are very similar to methanol synthesis catalysts, with mechanisms that are supposed to be strongly related [260]. As said, the real catalysts, with copper contents that are usually > 50 wt%, are essentially constituted by unsupported copper particles modified by zinc species, likely in the form of ZnO or partially reduced $ZnO_x$ (x < 1) species strongly interacting with it [41,218]. Most of the ZnO-$Al_2O_3$ phase acts as a stabilizing agent for copper metal particles against sintering. Zinc-modified copper particles appear to represent the real active phase for both WGS and methanol synthesis from syngas, with a possible common role of formate species as surface intermediates [33]. It is worth noting that pure copper surfaces very weakly adsorb both CO and hydrogen, as well as $CO_2$ (Table 2), while the addition of the Zn species strongly increases its activity [261]. The mechanism of the WGS reaction over Cu/ZnO can be similar to that described above for $ZnAl_2O_4$ (Reactions (11) and (12)), with a likely activating effect of $Cu^+$ species which is known to adsorb CO strongly, thus favoring the formation of formate species on $ZnO_x$, and suggest a role of metallic copper favoring hydrogen recombination and desorption. On the other hand, the hydrogenations of both CO and $CO_2$ are certainly favored by the activation of hydrogen on the Cu-ZnO system where $H_2$ dissociation would occur on metallic copper, with the formation of Zn hydride species by some kind of spillover [218]. Such species may react with $CO_2$-producing formate ions that can either decompose to CO + $^-$OH at a lower hydrogen pressure (rWGS) or be further hydrogenated to methoxy

groups and finally methanol at a higher hydrogen pressure (methanol synthesis from $CO_2$). CO hydrogenation by hydride species can occur through the formation of formyl species (formally $^-CHO$) [262] that can be hydrogenated to methoxy groups directly or through formate species too. The faster reaction of $CO_2$ with respect to CO in methanol synthesis can be associated with the strong nucleophilicity of the carbon atom of $CO_2$ (much stronger than that of CO), possibly further increased by its previous interaction with cations through oxygen atoms, coupled with the strong electrophilicity of the hydride species of Zn-H. On the other hand, the promotional effect of CO in the hydrogenation of $CO_2$ over these catalysts can be associated with the occurrence of the WGS reaction with water produced by methanol synthesis from $CO_2$ [260].

It is clear that dissociative adsorption neither of CO nor of $CO_2$ (which, in fact, do not occur on copper and the other components of these catalysts) is needed for methanol synthesis. Additionally, the activation of hydrogen as nucleophilic hydride ions can be the key for Cu/ZnO-based systems that, in fact, are used for the hydrogenation of electrophilic carbon-containing molecules, such as $CO_2$ and ketones [7]. Finally, it is worth noticing that, although commercial LTWGS catalysts and methanol synthesis catalysts are mainly based on metallic copper, it seems likely that catalysis for these reactions is non-metallic, mainly based on metal-supported surface Cu-Zn oxide species.

The production of hydrocarbons from $CO_x$ hydrogenation occurs on metals that more strongly adsorb, but not too much, carbon monoxide and that also easily adsorb hydrogen [148]. These metals are reported to be active in CO dissociation, producing carbon species or metal carbides, as well as in $CO_2$ dissociation. They can catalyze hydrocarbon synthesis both as bulk metals and when supported. However, while the chain growth mechanism is very efficient on cobalt and iron catalysts, it is less for nickel catalysts [263], which, in fact, are used for methanation but are less efficient for Fischer-Tropsch due to limited chain growth.

In the case of nickel catalysts for methanation and cobalt catalysts for LTFTS, ionic oxides act as the best carriers, alumina seeming to be the choice support. Other ionic oxide components are usually present too, acting as activators. The amount of metal loaded is similar to that allowing the full coverage of the support surface, finally producing very small metal particles strongly interacting with the support. This is a key factor in reducing the probability of producing volatile metal carbonyl molecules (in particular, $Ni(CO)_4$ in the case of nickel catalysts) and also in limiting the formation of carbon and the carburization of the metal. In fact, data suggest that carbon species preferentially form on large metal particles.

Studies provide evidence of the possible role of the support in hosting and activating surface-oxidized intermediates such as formate and methoxy species over both cobalt/alumina and nickel/alumina catalysts [264], with a possible role of hydrogen spillover from the metal to the supporting oxide and a oxygenate mechanism occurring on the support surface for both methanation and FTS. However, mechanisms via oxygenate intermediates (carboxylate, carbonate, formate, formyl, and methoxy) could also occur at nickel surfaces [265,266]. On the other hand, the availability of the support surface in the case of this system is very small, being the surface covered by dispersed metal particles. On the other hand, most of the data strongly converge for a predominant role of CO dissociation over the metal surface and carbide mechanisms for both reactions when carried out with syngas as a reactant [100]. Thus, the main role of the ionic support for Ni and Co catalysts for syngas conversion is to produce and stabilize very small metal particles resistant to sintering and carbon deactivation.

Nevertheless, the relatively strong adsorption of $CO_2$ over ionic oxides used as catalyst carriers, e.g., on aluminas doped with components with a basic character, such as K-doped alumina [267], $MgO-Al_2O_3$ [268] and $La_2O_3/Al_2O_3$ [269] may have a role in $CO_2$ methanation. In fact, a positive effect of support basicity on $CO_2$ methanation on supported nickel has been reported [130].

## 8. Conclusions

The data summarized in this review paper provide examples of how complex the science of catalysis and catalyst engineering are, even for reactions realized with the same or related reactants and in quite similar conditions. Oxide-supported metal catalysis, metal-supported oxide catalysis, as well as pure metal and pure oxide catalysis, and carbide catalysis as well, are involved in the conversion of syngases and the hydrogenations of carbon oxides. The data concerning catalysis in practical conditions can be successfully interpreted by taking into account surface chemistry and physical chemistry studies and DFT calculations, showing the usefulness of surface physics and chemistry studies. It is also clear that the cost of the catalyst may also have some relevance and that the use of the very expensive platinum group and noble metals is avoided when increases in performance are not very relevant.

The metals which play a key role in the reactions considered here are those that adsorb hydrogen and carbon monoxide with intermediate strength, in agreement with the Sabatier principle. On the other hand, the industrial catalysts for the reactions considered here are first-row transition metals (Cu, Ni, Co, and Fe), which are far cheaper than platinum group metals because, for these reactions, these metals are sufficiently competitive. Among these metals, those that more strongly adsorb CO are more performant in the Fischer-Tropsch reaction, where CO dissociation certainly plays a central role. On the other hand, oxide supports also have a main role from different points of view. The main role of support is to optimize the size and shape of the supported metallic particles, particularly in the case of alumina as the support of nickel catalysts for methanation and cobalt catalysts for LTFT, probably with the result of limiting the tendency to carburization of the corresponding metals. As for the reactions that do not imply the full breaking of the C-O bond of carbon oxides, the need for strong CO adsorption vanishes. On the other hand, we have emphasized that, although the metallic nature of copper is likely useful to provide hydrogen easy adsorption, the activity of methanol synthesis and LTWGS catalysts, both based on Cu-ZnO is (strictly speaking) not metallic. In fact, these catalysts are essentially constituted by unsupported copper modified by surface $ZnO_x$ species and likely work as copper-supported Cu-Zn oxides. In particular, the weaker interaction of CO with cationic centers and the more nucleophilic character of surface hydride species may be the key features for catalyzing these reactions. On the other hand, the HTWGS reaction catalysis implies fully oxidic catalysts such as those based on $Fe_3O_4$ and those based on $ZnAl_2O_4$.

It is clear that the catalytic conversion of captured $CO_2$ with green hydrogen to hydrocarbons and alcohols (methanol) is a possible method of producing sustainable e-fuels, such as e-methanol, e-methane, e-gasoline, e-jet fuel, and e-Diesel fuel. The development of optimized catalysts for these reactions is already at a nearly industrial level, and this allows the development of reactors and the corresponding manufacturing processes. According to the data discussed above, a slight modification of catalysts used for syngas conversion is apparently a successful approach for developing new $CO_2$-based processes. A different situation is that of reverse WGS processes. In fact, this reaction, although strictly related to direct WGS processes, needs much higher temperatures to be realized in significant amounts. Thus, very different catalysts practically act for rWGS with particular respect to LTWGS.

Although, as said, the state of the art seems to indicate already clear ways to optimize the new processes for the production of e-fuels and renewable chemicals from $CO_2$ hydrogenation, extensive research is still needed to clarify the details of the mechanisms of these reactions and the roles of the different catalyst' components, which are really still incompletely known. This work is likely needed to further improve the catalytic systems and might also allow us to discover more efficient materials.

In any case, it is clear that heterogeneous catalysis will be a key tool for the production of sustainable fuels, particularly e-fuels, just as it has been in the ending era of fossil hydrocarbon industrial chemistry.

**Author Contributions:** G.B.: conceptualization, validation, writing—review and editing; E.S.: thermodynamic calculations, conceptualization, validation; P.R.: conceptualization, validation, writing—review and editing; G.G.: funding acquisition, conceptualization, validation, writing—review and editing. All authors have read and agreed to the published version of the manuscript.

**Funding:** G.B., P.R. and G.G. acknowledge the PROMETH2eus project, PNRR project of MASE italian ministry (missione 2 "Rivoluzione verde e transizione ecologica", componente 2 "Energia rinnovabile, idrogeno, rete e mobilità sostenibile", investimento 3.5 "Ricerca e sviluppo sull'idrogeno"). E.S. acknowledges National Recovery and Resilience Plan (NRRP), Mission 4 Component 2 Investment 1.3—Call for tender No. 1561 of 11.10.2022 of Ministero dell'Uni-versità e della Ricerca (MUR); funded by the European Union—NextGenerationEU Project title "Network 4 Energy Sustainable Transition—NEST" (project code: PE0000021).

**Data Availability Statement:** The data in this review paper are from the cited references.

**Conflicts of Interest:** The authors declare no conflicts of interest.

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
