# Peer review of "Mechanistic and Compositional Aspects of Industrial Catalysts for Selective CO2 Hydrogenation Processes"

_catalysts, doi:10.3390/catal14020095_

Round 1
Reviewer 1 Report
Comments and Suggestions for Authors
In this manuscript, Busca et al. review the catalytic conversion of CO2 through classical heterogeneous processes such as WGS, CO2 to methanol, CO2 methanation, FTS and so on. The review is potentially interesting and, at some points, insightful, however, my feeling is that it is still far from being publishable for various issues detailed below:
a) The review goes through several topics of CO2 conversion, each of which, has been reviewed infinite times individually (the authors should mention previous review and describe why they believe this review shows innovative concepts). It may make some sense to review these topics together if one identifies and discusses in detail common mechanistic features and differences that can be helpful for a better understanding of catalysis science. At the moment the topics are not interconnected and I do not see the point of reviewing them together.
b) The authors managed to prepare a whole review without even adding a single Figure (or at least I did not find them). This is quite new to me. In general reviews of this kind are made to rationalize reaction mechanisms, trends and connections between different area of research even with help of graphics and schemes. I strongly recommend that the authors add at least a general scheme to show the reaction network and several mechanistic schemes.
c) The review contains too much description of industrial catalysts and their brand names and too less mechanistic insights. At least the authors should present tables comparing the performance of different catalysts (CO2 conversion, time-on-stream, etc.) and try to explain their mechanistic differences and which aspects make them more/less active.
d) The conclusions are just a few lines. In general, in this kind of overviews the conclusions are the crucial aspect where the authors provide their personal insight on which catalysts should be prepared in the future and what are the crucial developments to expect. The conclusions should occupy several pages of insightful discussion.
Comments on the Quality of English LanguageThe English is ok, some expressions are not very clear such as
abstract "The difference between conventional catalysts and those needed for pure CO2 conversion is emphasized." apart that the English is not clear, I do not feel that the authors actually provided such emphasis
Line 107, something with a segnalibro non definito....(seems like a Microsoft error on automatic numbering)
Line 217 "....while also the water gas shift equilibrium (reaction (1)) may be reached. "? reached where?
Line 231 "occidental countries" from "occidentale" in Italian, but it would be better to say "Western countries"
Author Response
In this manuscript, Busca et al. review the catalytic conversion of CO2 through classical heterogeneous processes such as WGS, CO2 to methanol, CO2 methanation, FTS and so on. The review is potentially interesting and, at some points, insightful, however, my feeling is that it is still far from being publishable for various issues detailed below:
- The review goes through several topics of CO2 conversion, each of which, has been reviewed infinite times individually (the authors should mention previous review and describe why they believe this review shows innovative concepts). It may make some sense to review these topics together if one identifies and discusses in detail common mechanistic features and differences that can be helpful for a better understanding of catalysis science. At the moment the topics are not interconnected and I do not see the point of reviewing them together.
The reviewer is correct underlining that many reviews cover the reactions considered here. These reviews usually report on laboratory studies with infinite lists of different catalyst compositions tested, including essentially compounds of the entire element table. On the other hand, although academic researchers correctly insist on studying different catalyst compositions, the industry uses for decades only a few of them (most frequently essentially one for each reaction) that are well established to be by far the most performant and stable. Here the characteristics of the most performant, and consequently commercial, catalysts for different but related reactions are comparatively discussed. We relate the behavior of the most performant industrial catalysts for different but interconnected reactions, each other, and we base the discussion on the specific surface chemistry of the different catalyst components. We do not find any other paper/review written with this approach in the literature.
This approach is quite innovative, and useful, setting the fundamental state of the art from relevant conditions applied at the industrial level. As an example, we underline here that, although methanol synthesis catalysts are generally reported to be based on ZnO-supported copper, this is absolutely false. Copper is certainly not supported on ZnO, but, very likely, the reverse is true: methanol synthesis catalysts are based on Cu-supported ZnO species.
- The authors managed to prepare a whole review without even adding a single Figure (or at least I did not find them). This is quite new to me. In general reviews of this kind are made to rationalize reaction mechanisms, trends and connections between different area of research even with help of graphics and schemes. I strongly recommend that the authors add at least a general scheme to show the reaction network and several mechanistic schemes.
We added five figures to try to explain better some points in the review.
- The review contains too much description of industrial catalysts and their brand names and too less mechanistic insights. At least the authors should present tables comparing the performance of different catalysts (CO2 conversion, time-on-stream, etc.) and try to explain their mechanistic differences and which aspects make them more/less active.
As said above, evident in the title, and explained better in the revised text, this review focuses on industrial catalysts: they are the only one really applied and, obviously, the most performant known today. We would like to underline that this work contributes in setting the ground target when developing new materials.
- The conclusions are just a few lines. In general, in this kind of overviews the conclusions are the crucial aspect where the authors provide their personal insight on which catalysts should be prepared in the future and what are the crucial developments to expect. The conclusions should occupy several pages of insightful discussion.
Most of the deepness of this review paper is in chapters 5 and 6. We enlarged a little bit the conclusions, but we do not want to repeat concepts.
Comments on the Quality of English Language
The English is ok, some expressions are not very clear such as
abstract "The difference between conventional catalysts and those needed for pure CO2 conversion is emphasized." apart that the English is not clear, I do not feel that the authors actually provided such emphasis
Modified
Line 107, something with a segnalibro non definito....(seems like a Microsoft error on automatic numbering)
Corrected
Line 217 "....while also the water gas shift equilibrium (reaction (1)) may be reached. "? reached where?
Modified
Line 231 "occidental countries" from "occidentale" in Italian, but it would be better to say "Western countries"
Modified
Reviewer 2 Report
Comments and Suggestions for Authors
In the present work, the authors studied “Mechanistic and compositional aspects of industrial catalysts for selective CO2 hydrogenation processes.”. The manuscript needs to be improved in many sections. As a reviewer, I would suggest that this manuscript can be accepted after a major review.
The abstract needs to be changed in a way that the novelty of the article. As far as I can see they are just repeating the same sentences with different words.
The introduction is not enough need to increase in a way that the readers should be more curious about the outcome of the article.
References should be added to the tables.
Only one catalyst for the one process why? In a review, you should take more examples.
The graphs should be added to make the article more interesting.
For each process, the theoretical calculations of Gibbs free energy should be presented in graphs.
Finally, conclusions need to be changed.
Comments on the Quality of English LanguageEnglish correction is needed throughout the manuscript.
Author Response
In the present work, the authors studied “Mechanistic and compositional aspects of industrial catalysts for selective CO2 hydrogenation processes.”. The manuscript needs to be improved in many sections. As a reviewer, I would suggest that this manuscript can be accepted after a major review.
The abstract needs to be changed in a way that the novelty of the article. As far as I can see they are just repeating the same sentences with different words.
We slightly modofied the abstract
The introduction is not enough need to increase in a way that the readers should be more curious about the outcome of the article.
We modified the introduction
References should be added to the tables.
Only one catalyst for the one process why? In a review, you should take more examples.
The commercial catalysts, those actually used by industry, belong to a single catalyst system for each reaction. This is because, after decades of research and industrial application, the most effective composition is the only really used.
The graphs should be added to make the article more interesting.
We added five figures
For each process, the theoretical calculations of Gibbs free energy should be presented in graphs.
We added the calculation of standard free energy change for WGS, methanations and methanol syntheses
Finally, conclusions need to be changed.
Some modification of the conclusions section was done
Round 2
Reviewer 1 Report
Comments and Suggestions for Authors
I went through the revised version of this review and I found it improved to the level that may be acceptable in Catalysts after some minor revisions:
a) The authors have improved the introduction by adding emphasis to the main concept of this work (review of industrial catalysts and their composition). In this context, they have added a new paragraph to the introduction (lines 46-64) almost without references. As the authors refer to "an enormous number of different catalyst compositions are object of investigation in academic research", it could be useful to add some references at this point.
b) The authors have added a few schematic figures to the review which is ok but they could provide a more detailed explanation in the captions. Moreover, in some cases, I guess (Figure 3 for examples) the figures are taken from published manuscript but the mention is missing in the caption. In Figure 4, the meaning of BFW should be explained, etc.
c) The authors mention in their response that "Most of the deepness of this review paper is in chapters 5 and 6. We enlarged a little bit the conclusions, but we do not want to repeat concepts." It is strange to receive such a comment by an author of such expertise or, perhaps my comment was not clear. The conclusions of a review are surely not used to repeat the content of previous sections but they should contain an outlook and perspective that, based, on the content of the manuscript, will drive future endeavors. To make things clearer my recommendation is that the authors prepare an "Outlook and perspective" section where they comment on future possible development of the field so that the conclusion, that they have already added to the text, can be used just to summarize the main aspects of work.
Author Response
I went through the revised version of this review and I found it improved to the level that may be acceptable in Catalysts after some minor revisions:
- The authors have improved the introduction by adding emphasis to the main concept of this work (review of industrial catalysts and their composition). In this context, they have added a new paragraph to the introduction (lines 46-64) almost without references. As the authors refer to "an enormous number of different catalyst compositions are object of investigation in academic research", it could be useful to add some references at this point.
Taking into account that we are speaking about several reactions, all of which were under deep investigations, several dozens of reviews have been recently published on these subjects. Thus we opted to cite an example, in the reviewed paper.
- The authors have added a few schematic figures to the review which is ok but they could provide a more detailed explanation in the captions. Moreover, in some cases, I guess (Figure 3 for examples) the figures are taken from published manuscript but the mention is missing in the caption. In Figure 4, the meaning of BFW should be explained, etc
The figures are original artwork derived from the literature, which is cited in the caption. In Fig. 4 the meaning of BFW is now reported
- The authors mention in their response that "Most of the deepness of this review paper is in chapters 5 and 6. We enlarged a little bit the conclusions, but we do not want to repeat concepts." It is strange to receive such a comment by an author of such expertise or, perhaps my comment was not clear. The conclusions of a review are surely not used to repeat the content of previous sections but they should contain an outlook and perspective that, based, on the content of the manuscript, will drive future endeavors. To make things clearer my recommendation is that the authors prepare an "Outlook and perspective" section where they comment on future possible development of the field so that the conclusion, that they have already added to the text, can be used just to summarize the main aspects of work.
We added few additional words in the conclusions to underline that research is needed to optimize CO2 hydrogenation processes, and, in particular, to better understand mechanistic aspects.

Reviewer 2 Report
Comments and Suggestions for Authors
In the present work, the authors studied “Mechanistic and compositional aspects of industrial catalysts for selective CO2 hydrogenation processes.” The revised version of the manuscript is greatly improved in many sections that are pointed out by reviewers. As a reviewer, I would suggest that this manuscript is acceptable in its present form.
Finally, I would like to suggest authors make thorough language corrections.
Author Response
n the present work, the authors studied “Mechanistic and compositional aspects of industrial catalysts for selective CO2 hydrogenation processes.” The revised version of the manuscript is greatly improved in many sections that are pointed out by reviewers. As a reviewer, I would suggest that this manuscript is acceptable in its present form.
Finally, I would like to suggest authors make thorough language corrections.
We correcte several misprints and modified few statements